# 4polar3D single molecule imaging of 3D orientation in dense actin networks using ratiometric polarization splitting

Charitra S. Senthil Kumar [1,7], Cesar A. Valades Cruz [2,7], Miguel Sison[1,7], Arturo G. Vesga [3], Javier Rey-Barroso[4,6], Valentina Curcio[1], Luis A. Alemán-Castañeda [1], Miguel A. Alonso[1,5], Renaud Poincloux [4], Manos Mavrakis [1] & Sophie Brasselet [1] ✉

Single Molecule Orientation and Localization Microscopy (SMOLM) aims at simultaneously measuring the position and orientation of single molecules, generating orientation-encoded super-resolved images by estimating both their 3D mean orientation and the extent of their angular fluctuations (wobble). Most existing SMOLM approaches rely on the engineering of single molecules' point spread functions, which requires complex optical setups and long computational times that can be an obstacle in dense cellular environments with high detection density and challenging imaging conditions. In this work, we propose a simpler and effective method named 4polar3D, based on the estimation of single molecule intensities projected onto four polarized channels with controlled numerical apertures. This strategy enables 3D orientation measurements of single molecules within a 0-180° azimuthal range in addition to their angular range of fluctuations and their 2D localization, using a setup requiring minimal alignment complexity. It is moreover based on pure intensity-estimation, making data processing considerably faster than complex PSF shape analysis and relatively insensitive to geometrical aberrations. We demonstrate that 4polar3D can resolve nanoscale molecular organization in whole cells' crowded structures, uncovering 3D-oriented actin filament networks in densely packed lamellipodia and podosomes.

Single Molecule Orientation and Localization Microscopy (SMOLM) emerged a decade ago with the aim of achieving localization and orientation imaging of isolated fluorescent molecular dipoles[1,2]. Resolving single molecules' (SMs) orientation in addition to their spatial position enables the investigation of the nanoscale 3D organization of proteins, which is crucial to understanding a wide range of biological processes from immunology to mechanobiology. Molecular orientation plays indeed a critical role in biological function, not only at the level of proteins' conformational changes, but also in the context of their collective structural organization. To probe the orientational organization of proteins in a cell, standard single-molecule localization microscopy (SMLM) is often not enough. For instance, actin filaments in the cell cytoskeleton are so dense that such organization is not visible within the packed collection of localizations. Therefore, specific

[1]Aix Marseille Univ, CNRS, Centrale Med, Institut Fresnel, Marseille, France. [2]Institute of Hydrobiology, Chinese Academy of Sciences, Wuhan, China. [3]Optical Cell Biology Group, Gulbenkian Institute for Molecular Medicine, Lisbon, Portugal. [4]Institut de Pharmacologie et de Biologie Structurale (IPBS), Université de Toulouse, Toulouse, France. [5]The Institute of Optics, University of Rochester, Rochester, NY, USA. [6]Present address: Faculty of Experimental Sciences, Universidad Francisco de Vitoria, Madrid, Spain. [7]These authors contributed equally: Charitra S. Senthil Kumar, Cesar A. Valades Cruz, Miguel Sison. ✉e-mail: sophie.brasselet@fresnel.fr

orientation-sensitive methods need to be developed. Since the point spread function (PSF) of single molecules imaged with a high numerical aperture (NA) microscope encodes information on their orientation, most SMOLM strategies rely on analyzing the shape of engineered PSFs, typically modified using phase or polarization masks placed at the pupil plane of the microscope objective[3–9]. PSF engineering enables precise estimation of both the average orientation of SMs as well as their rotational dynamics referred to as wobble, which reflects the extent of their angular fluctuations over tens of milliseconds. This approach requires, however, advanced optimization and retrieval strategies[2], such as PSF basis decomposition[6,10], maximum likelihood estimation[11–13] or deep learning approaches[14,15]. Despite the potential of PSF engineering for SMOLM, it has been predominantly applied to in vitro systems[11,16,17] rather than to complex, densely packed cellular environments, which typically require a very high number of localizations over large fields of view. PSF engineering approaches have several inherent limitations, namely the complexity of microscope setups to be implemented, the use of enlarged PSFs that decrease the accessible detection density, the need to correct for phase and polarization aberrations[18–21], and the substantial computational complexity associated with parameters' retrieval, often resulting in slow data analysis procedures.

An alternative strategy to PSF engineering relies on the ratiometric intensity estimation of the signal from SMs polarized into different polarization channels, in addition to standard 2D localization. By projecting the fluorescence signal onto two or four polarization channels, it is possible to estimate SMs' anisotropy in isotropic environments[22] or to determine their 2D orientation projected in the sample plane[23–26]. Extending this approach to 3D orientation estimation is not straightforward, as polarization is generally manipulated in the transverse optical plane to the propagation direction. So far, 3D orientation determination using ratiometric polarization-based methods has been achieved only indirectly, supposing that the molecules are fixed or have a known wobble[27,28], involving sequential measurements with multiple directions of illumination[29], or fitting polarized PSFs[13,26,30] at the price of an increase in computation complexity. Among the least computationally demanding strategies, ratiometric polarization-based estimation seems highly appropriate for its relatively low sensitivity to microscope's optical aberrations, its decrease in processing time of SMOLM, and its capacity to access information from large data sets in crowded environments.

In this work, we propose a simple, yet powerful scheme for measuring 3D orientation and wobble of SMs using a four-polarization ratiometric approach that relies solely on intensity estimations across distinct polarization channels. This scheme combines polarization splitting to extract in-plane orientation information with selective detection in the pupil plane of the microscope, to access off-plane orientation information. The approach capitalizes on the fact that the angular emission pattern radiated by a molecular dipole is stronlgy dependent on its off-plane tilt angle[31]. Specifically, when the dipole is tilted off-plane (i.e., perpendicular to the sample plane), the border region of the pupil plane, corresponding to the most tilted radiation wave vectors, is brighter than the central region (Fig. 1a, left). Inversely, when the dipole lies in the sample plane, the central region is brighter (Fig. 1a, right).

To achieve 3D orientation estimation, the method thus combines four polarization splitting channels with different effective numerical apertures (NA s). First, the signal is split onto two channels of different NA s, respectively low NA ($NA_{low}$) and high NA ($NA_{high}$) using two different diaphragms, D1 and D2, respectively (Fig. 1b, c). Each of these two unpolarized channels is then split into two linear polarizations, 0°/90° and 45°/135° respectively, with respect to the horizontal axis of the sample plane[25] in a relatively compact and simple implementation (Fig. 1b). This method, named 4polar3D, combines the benefits of transverse 4-polarization splitting and pupil amplitude splitting (e.g.

NA filtering) to determine the azimuth angle $\xi$ (i.e., in-plane orientation within the 0-180° range), polar angle η (i.e. off-plane orientation), and the wobble angle $\delta$ of SMs (Fig. 1d, inset), from ratiometric intensity measurements. This approach is particularly advantageous for its insensitivity to optical aberrations and its compatibility with relatively high-density images, which can be detrimental in PSF engineering techniques (see Supplementary Note 1). We apply this method in dense actin filament assemblies in cells, namely in the lamellipodia of motile cells and in podosomes, whose 3D organization architecture is complex and so far only accessible using electron tomography.

## Results
### Principle and data processing
The optical setup is based on a simple split of the image plane into 4 polarization channels referred to as 0°, 90°, 45° and 135°. The polarization quality of both excitation and detection polarizations is ensured by compensation methods (see Methods). The numerical apertures of the polarized channels, $NA_{low}$ and $NA_{high}$, are adjusted by diaphragms placed in the relayed pupil planes (Fig. 1a–c) (see Materials and Methods). $NA_{high}$ is chosen to lie below the critical angle aperture to avoid coupling effects with super-critical angle fluorescence (SAF) emission at the proximity to the glass coverslip interface. Avoiding SAF also ensures that the total energy collected by a dipole is almost identical whether it is oriented in-plane or along the axial direction[32]. Typically, $NA_{high} \sim 1.3$ (corresponding to the critical angle aperture for a glass-water interface), which corresponds to about 60% of the total signal remaining detected for an unpolarized emitter. $NA_{low}$, on the other hand, is large enough to allow a sufficient number of photons to be detected and small enough to allow sensitivity to 3D orientations. This corresponds to about 80% of the high NA ($NA_{low} \sim 1.1$).

The analysis of the 4polar3D approach is based on the relative comparison of the four intensities ($I_0, I_{90}, I_{45}, I_{135}$), integrated over the PSF of each single molecule, such as displayed in Fig. 1c (see Supplementary Note 1 for a description of the detailed model). The relative integrated intensities of these PSFs depend on the molecular orientation parameters, described by the molecular dipole mean 3D direction ($\eta, \xi$), and its wobble extent $\delta$ around this direction (Fig. 1d), averaged over the integration time of the camera. We suppose that molecules are photoexcited almost isotropically (i.e. for instance using circularly polarized total internal reflection), and that their rotational diffusion is fast compared to the camera integration time. While polarized PSFs may exhibit non trivial shapes, especially for fixed and highly tilted molecules (Supplementary Fig. S1), their size is limited to the diffraction limit, in contrast to PSF engineering, which expands the PSF size to at least twice this limit. This feature offers a strong advantage in situations of high detection densities. The estimation of the integrated intensities ($I_0, I_{90}, I_{45}, I_{135}$) from single molecule PSFs is addressed in the section below, dedicated to the parameters' retrieval. Supposing that these intensities are properly estimated, their dependence on the molecular angular parameters ($\eta, \xi, \delta$) depicted in Fig. 1d–f (Supplementary Note 1) shows that simple combinations of ($I_0, I_{90}, I_{45}, I_{135}$) provide unambiguous sensitivity to these parameters. First, because of the dependence of the intensity pattern on off-plane dipole orientation (Fig. 1c), the ratio between the high-NA and low-NA channels ($I_0 + I_{90}$)/($I_{45} + I_{135}$), is sensitive to the off-plane orientation angle η, with a higher sensitivity at low wobble angles $\delta$, as expected for fixed dipoles[31] (Fig. 1d). Second, we notice that in the paraxial regime, the ratios $P_0 = (I_0 - I_{90})/(I_0 + I_{90})$ and $P_{45} = (I_{45} - I_{135})/(I_{45} + I_{135})$ can be directly related to the normalized Stokes parameters, from which 2D orientation $\xi$ and wobble $\delta$ can be determined through simple mathematical operations (Supplementary Note 1)[25]. In particular, atan($P_{45}/P_0$) is sensitive to $\xi$ (Fig. 1e), while the norm $P_0^2 + P_{45}^2$ depends on $\delta$ (Fig. 1f). These dependencies show that this scheme is capable of differentiating 3D orientation from wobble, which is not

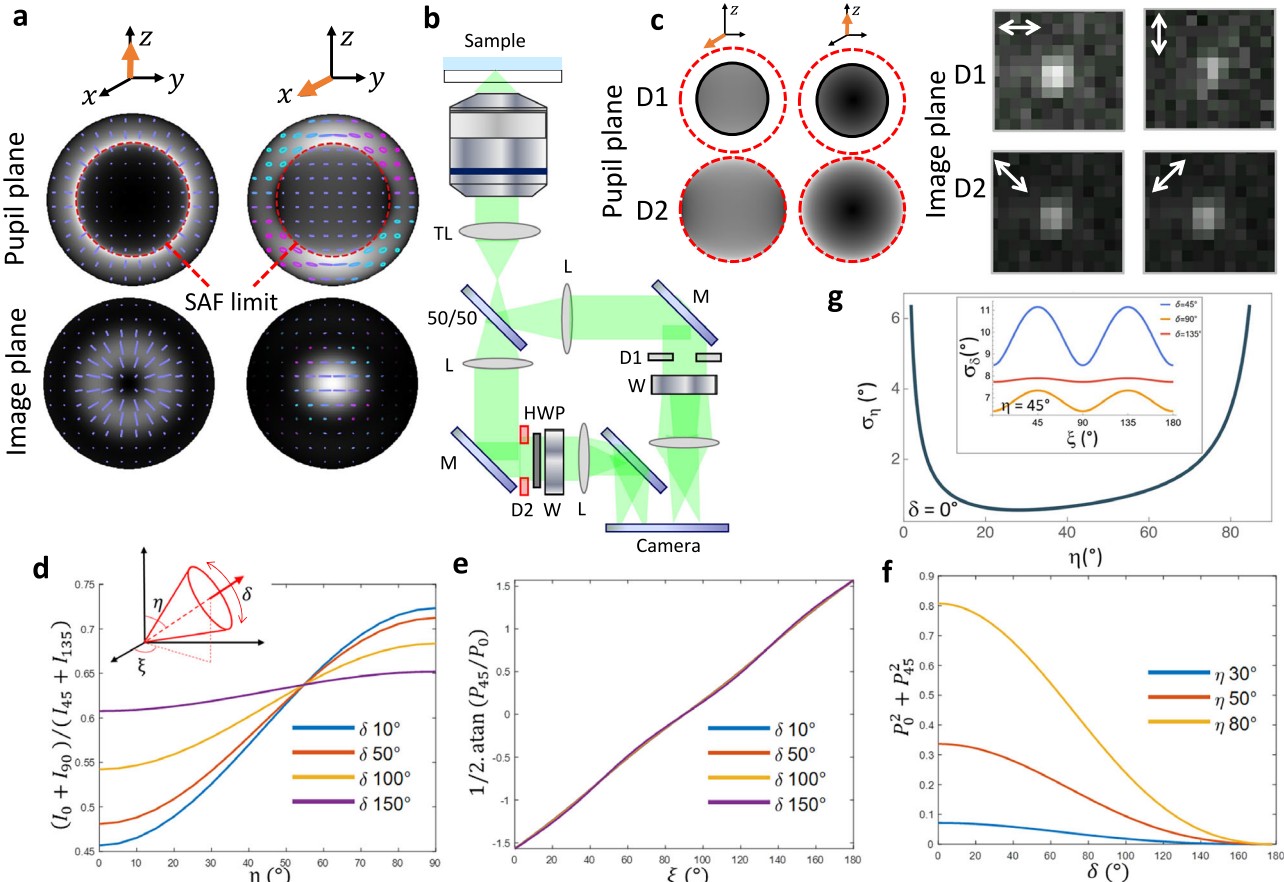

**Fig. 1 | Principle of 4polar3D. a** Intensity and polarization distributions in the pupil plane (top) and image plane (bottom) of a dipole (orange arrow) oriented along the longitudinal axis z (left) and the in-plane axis x (right). The intensity pattern (black and white image) is used as a background and the local polarization is drawn with a shape and color code representing its ellipticity (cyan: left handedness, magenta: right handedness). The dashed red circle corresponds to the super critical angle emission. **b** Schematic representation of the 4polar3D setup. TL: tube lens, 50/50: non-polarizing beam splitter, L: lenses, M: mirrors, D1 and D2: diaphragms, W: Wollaston prism, HWP: half wave plate. **c** Left: theoretical intensity distributions in the pupil plane for an in-plane dipole (left) and off-plane dipole (right), in the channels measured respectively at $NA_{low}$ (diaphragm D1, top) and $NA_{high}$ (diaphragm D2, bottom). Right: example of experimental PSFs from a single molecule along the four polarized channels (represented by the white arrow) (pixel size 130 nm). The (0°,90°) and (45°,135°) channels are measured respectively at $NA_{low} = 1.1$ (diaphragm D1) and $NA_{high} = 1.3$ (diaphragm D2). **d** Dependence of the intensity ratio $(I_0 + I_{90})/(I_{45} + I_{135})$ on $\eta$ for different values of $\delta$. Inset: definition of the angles $\xi, \eta, \delta$, the red arrow defines the mean direction of the single molecule wobbling cone. **e** Dependence of $1/2.\mathrm{atan} P_{45}/P_0$ on $\xi$ for different values of $\delta$. **f** Dependence of $P_0^2 + P_{45}^2$ on $\delta$ for different values of $\eta$. **g** Cramér Rao lower bound on the expectation of the $\eta$ parameter, for $\xi = 0°$ and a fixed dipole ($\delta = 0°$). Conditions used: total intensity 5000 photons, $n_0 = 1.33$, $n_1 = 1.515$, $z_0 = z_1 = 0$, $NA_{low} = 1.04, NA_{high} = 1.3$ (under critical angle condition, $NA$ ratio 0.8). Inset: CRLB on $\delta$ calculated as a function of $\xi$, for different values of $\delta$, for $\eta = 45°$.

the case in a 2D polarization splitting configuration[25]. Notably, these dependencies are also in theory robust to defocus and to the presence of unpolarized aberrations of the system, providing that numerical apertures stay under the critical angle (see Supplementary Note 1 and Supplementary Fig. S2). Note that an intrinsic property of the setup design is that $\xi$ is determined within the limit [0°–180°], and not [0°–360°], which is due to the circular geometry of the diaphragm used in the low- $NA$ channel, introducing an ambiguity between orientations symmetric by 180° rotation with respect to the longitudinal axis $z$. 4polar3D thus estimates an off-plane angle $\eta$ that is the amount of tilt of SMs relative to the sample plane.

The quantitative retrieval of the $(\eta, \xi, \delta)$ parameters from the four polarized intensities follows a simple procedure (see Supplementary Note 1). Polarized PSFs detected in the four polarized channels are linear functions of the second order moments $\boldsymbol{m}$ of the molecular dipole vector components $\mu_i$, expressed as the time averaged quantities $m_{ij} = \langle \mu_i \mu_j \rangle (\Omega)$ with $\Omega = (\eta, \xi, \delta)$[33]:

$$\boldsymbol{I}(\boldsymbol{r}, \Omega) = \boldsymbol{K}(\boldsymbol{r}) \cdot \boldsymbol{m}(\Omega) \qquad (1)$$

with $\boldsymbol{I}(r, \Omega) = (I_0, I_{90}, I_{45}, I_{135})^T$ being the polarized PSFs that are functions of the spatial coordinate $\boldsymbol{r}$ in the image plane. $\boldsymbol{K}(\boldsymbol{r})$ is a 4×4 propagation matrix that depends on the detection numerical apertures as explicited in Supplementary Note 1, and $\boldsymbol{m} = (m_{xx}, m_{yy}, m_{zz}, m_{xy})^T$ is the vector of second-order moments accessible in 4polar3D. We note that the cross-terms $(m_{xz}, m_{yz})$ are absent from the estimated vector $\boldsymbol{m}$, as a consequence of the use of centered circular apertures as mentioned above. In 4polar3D, the total intensities $\langle \boldsymbol{I} \rangle = \int \boldsymbol{I}(\boldsymbol{r}, \Omega) d\boldsymbol{r}$, integrated over the whole PSF, are used to retrieve the components of $\boldsymbol{m}$ from a simple inversion operation:

$$\boldsymbol{m}(\Omega) = \langle \boldsymbol{K} \rangle^{-1} . \langle \boldsymbol{I} \rangle \qquad (2)$$

The components of the integrated $\langle \boldsymbol{K} \rangle$ matrix are determined from calibration involving fluorescent nano-beads and a rotating polarizer (see Supplementary Note 2 and Supplementary Fig. S3), an effective way of mimicking depolarized or transverse dipoles[34]. Once $\boldsymbol{m}$ is estimated, the retrieval of the parameters $\Omega = (\eta, \xi, \delta)$ is performed using analytical expressions of $\boldsymbol{m}(\Omega)$, possibly supplemented by a minimization (Supplementary Note 1).

The sensitivity of the approach to $(\eta, \xi, \delta)$ in the presence of noise was assessed by computing the Cramér-Rao Lower Bound (CRLB) of the integrated intensities, using a typical situation ($NA_{high} = 1.3$, $NA_{low} = 1.1$) in the presence of Poisson noise and background (Supplementary Note 3). In a single-molecule regime of total intensity 5 000 photons (defined before any channel's split and $NA$ reduction) and in the presence of 10 background photons per pixel, the estimation of the angular parameters reaches a CRLB minimal standard deviation of about a few degrees for $(\eta, \xi)$ and less than 10° for $\delta$ (Supplementary Fig. S4). Regions of higher standard deviations can be observed for $\eta \sim 0°$ and $\delta \sim 180°$, which correspond to situations where $\xi$ is ill-defined. $\eta$ is also less precise for the extreme tilt angles ($\eta \sim 0°, 90°$) (Fig. 1g) as expected from the $\eta$-dependence of intensity ratios displayed in Fig. 1d. Note that under Poisson noise conditions, these standard deviations are expected to scale as $1/\sqrt{N}$, with $N$ the number of detected photons.

## Spatial localization and 3D orientation retrieval

The estimation of SM intensities relies on PSF shapes that are deformed by the polarization projection, especially for fixed orientations (Supplementary Fig. S1). In practice, 4polar3D employs a computationally simple procedure, in which we discard the option to run

multiple-parameter fitting algorithms that would require long optimizations. After the four polarized channels have been registered (see Supplementary Note 2), single molecules are detected in the four channels via a generalized likelihood ratio test algorithm[23,25] (Materials and Methods). An algorithm then estimates the intensity from each PSF. We tested three options: (1) fitting the PSFs with a simplistic, albeit erroneous Gaussian function model; (2) refining this fit to a 2-dimensional Gaussian, which is still computationally light and permits to account for most deformed PSFs; (3) blindly summing pixel intensity values inside a window around each detected PSF, after subtracting the background estimated from pixels around this area. Finally, the single molecule spatial 2D localization is extracted from the center of the Gaussian estimates, or the centroid in the case of the third option. To test the precision and accuracy of intensity estimation methods, the detection, estimation and retrieval steps are performed on synthetic images generated by a Monte Carlo simulation, where theoretical PSFs are affected by binning, noise and background in different situations of image degradation conditions (Supplementary Note 4). The retrieval of $(\eta, \xi, \delta)$ is then compared to their ground truth. Despite their deformation, the sizes of polarized PSFs cover a few camera pixels as in standard SMLM detection configurations (Fig. 2a). In the absence of noise, the determination of $(\eta, \xi, \delta)$ is

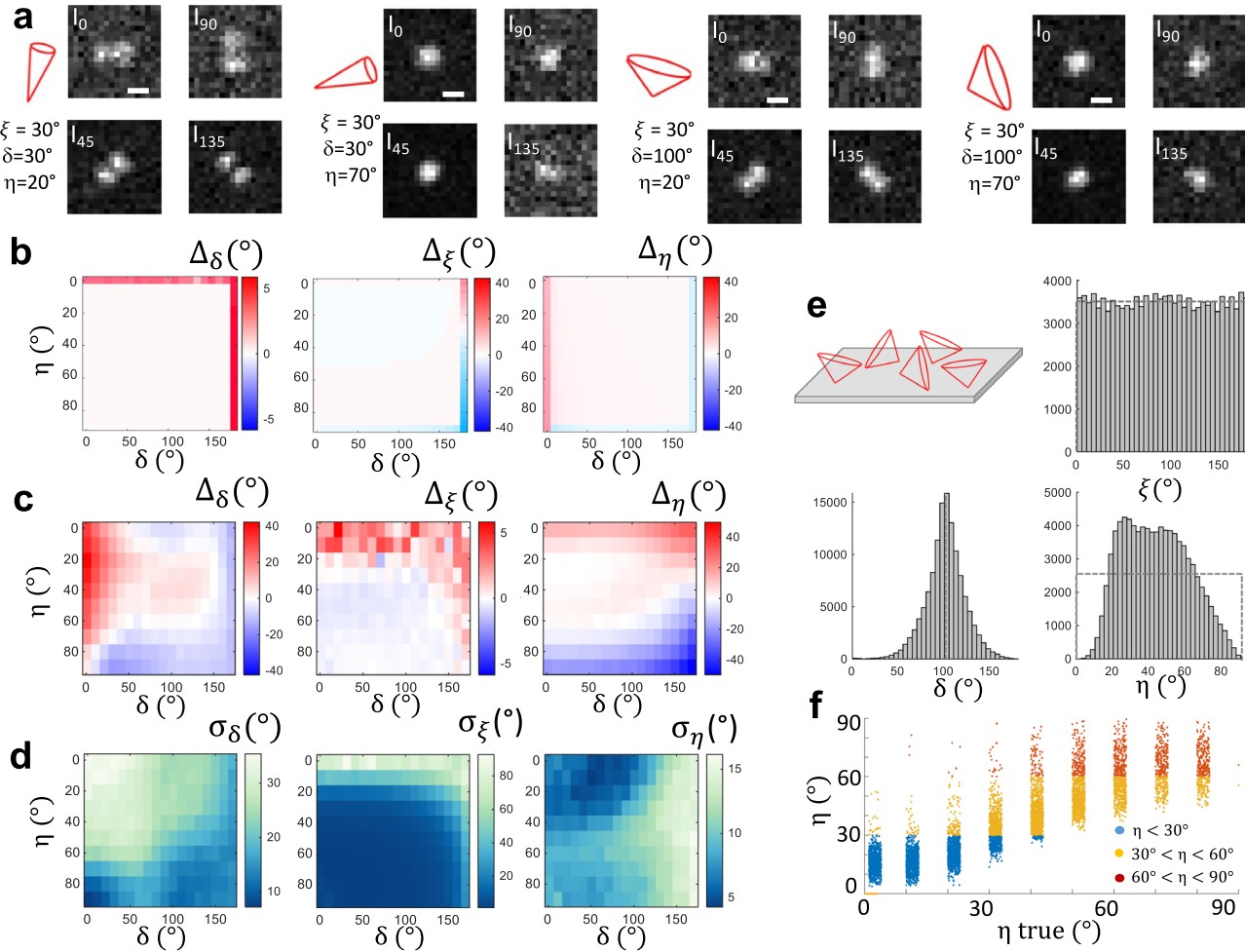

**Fig. 2 | Estimation performance of 4polar3D.** Conditions of Monte Carlo simulations: 10 molecules per image, 1000 images, 5000 photons/molecule, 10 ph/pixel background. **a** Examples of simulated PSFs (normalized to their maximum) in the presence of noise and background, with corresponding $(\xi, \eta, \delta)$ values indicated on the left of each image. Scale bar: 500 nm. **b** Accuracy of the parameters' retrieval in absence of noise. **c** Accuracy of the parameters' retrieval ($\Delta_\delta, \Delta_\xi, \Delta_\eta$ from left to right, values in degrees) in presence of noise, supposing the intensity estimated by

a Gaussian fit. **d** Precision of the parameters' retrieval ($\sigma_\delta, \sigma_\xi, \sigma_\eta$ from left to right, values in degrees). **e** Histograms of $\delta$, $\xi$ and $\eta$ obtained from a population of random values of $(\xi, \eta)$ for $\delta = 100°$. The ground truth distributions are indicated in dashed lines. The population considered is schematically represented as cones of random orientations distributed on a surface. **f** Retrieved $\eta$ values as functions of their ground truth value, color coded depending on the population they belong: off-plane [$\eta < 30°$], intermediate [$30° < \eta < 60°$], in-plane [$\eta > 60°$].

unbiased over the full range of parameter values, except for ill-defined cases mentioned above (Fig. 2b). This robustness under high signal to noise ratio (SNR) conditions has been previously exploited for the estimation of the 3D orientation of isolated single molecules under controlled orientation conditions[35]. To assess the effect of noise, background and spatial localization on the 4polar3D approach, we modeled a situation of molecules with a total intensity (defined before any channel's split and *NA* reduction) of 5 000 photons and a background of 10 photons/pixel per channel. Monte Carlo simulations show that high accuracy and precisions down to a few degrees can be reached in a range of off-plane angle and wobble $(\eta, \delta)$ around $20° < \eta < 70°$ and $30° < \delta < 150°$ (Fig. 2c, d), with a negligible dependence on the in-plane angle $\xi$. In this parameter range, the intensity estimation using Gaussian fits exhibits a better performance than the box-integration estimation (Supplementary Fig. S5), which might suffer from inaccurate estimation of the surrounding background. Surprisingly, the symmetric Gaussian and rotated asymmetric Gaussian fits exhibit very similar performances, showing that a traditional detection and intensity estimation from symmetric Gaussian shapes performs well despite the deformations of polarized PSFs. In the considered parameter range $(\eta \in [20° - 70°], \delta \in [30° - 150°])$ the precisions obtained from numerical simulations are close to the CRLB (Fig. 2d, Supplementary Figs. S4). At extreme values $(\eta \sim (0°, 90°), \delta \sim (0°, 180°))$ however, a loss of performance is observed, with a degradation that depends on the number of photons as expected (Supplementary Fig. S6). This originates from the relatively lower sensitivity of the ratiometric intensity quantities to these parameters in these ranges (Fig. 1d–g).

The consequence of the inaccuracy of the method for off-plane orientation limits is visible when simulating a collection of random orientations for a given wobble value $\delta$ (Fig. 2e). While the retrieval of $\delta$ and $\xi$ leads to expected distributions (Fig. 2e), the $\eta$ histogram is cropped at its extremes $\eta \sim (0°, 90°)$. This effect can be mitigated when considering binned $\eta$ values, opting for a data analysis that accounts for populations rather than a precise estimation of $\eta$ (typically off-plane $\eta < 30°$, intermediate $30° < \eta < 60°$, in-plane $\eta > 60°$). Figure 2f illustrates that by forming such binned populations of $\eta$, the portion of molecules belonging to the wrong population is, for $\delta = 100°$, 1% (for the off-plane situation), 15% (intermediate) and 23% (in-plane). Finally, the different estimation methods mentioned above were tested with respect to their performance in retrieving spatial 2D localization. The simulations show that the localization precision stays below 20 nm for all used methods at a signal condition of 5000 photons and a background of 10 photons/pixel per channel, with a localization bias below 15 nm (Supplementary Fig. S7). Note that the intrinsic localization bias due to the intensity-based nature of the estimation method is relatively low: at 10,000 photons (10 photons background/pixel), the expected $(x, y)$ localization precision is around 6 nm for most of the orientations (10 nm for off-plane molecules) and the localization bias is about 3 nm (8 nm for off-plane molecules) (Supplementary Fig. S7).

Overall, the estimation performance of 4Polar3D in orientation and spatial localization lies therefore in some aspects below the performance of the most efficient PSF engineering methods developed so far[1,2], for which precision and accuracy values have been found to reach a few degrees for all orientations and around 10° for wobble in similar signal to noise conditions[7,8]. In contrast, the strong advantages of 4Polar3D lie in the significant simplification of its implementation, as well as its simple and fast data processing. Finally, Monte Carlo simulations were used to estimate how geometrical aberrations and defocus affect the performance of 4polar3D. While in theory, phase perturbations in the pupil plane do not have any incidence on the measured intensities, aberrations induce PSFs deformations that may lead to errors in the polarized channels' intensity estimation under realistic noise conditions.

Supplementary Figs. S8, S9 show that excluding extreme off-plane orientations, the precision and accuracy of orientation parameters is not strongly affected by the presence of standard aberrations, while the accuracy of $\delta$ is degraded by about 10–20°.

## 4polar3D SMOLM in model lipid membranes

4Polar3D was implemented experimentally using a circular excitation polarization under total internal reflection fluorescence (TIRF) illumination, in order to balance the components of the excitation polarization over the three spatial axes (see Materials and Methods). 4polar3D was first validated on lipid membranes in which the mean orientation of embedded fluorophores is predictable (Fig. 3a). Supported lipid layers (SLB) made of 1,2-dipalmitoyl-sn-glycero-3-phospholine (DPPC) and cholesterol mixtures (DPPC:Chol 60:40 mole %) were labeled with Nile Red (NR) (see Materials and Methods), which transiently binds to lipid membranes[11]. Within the rigid lipid environment provided by DPPC:Chol, single NR dipoles tend to orient along the axial direction of the fatty acid chains[11], which is visible from their PSF shapes (Fig. 3b). A mean off-plane orientation of $\eta \sim 30°$ is measured on average, with a mean wobble angle of about $\delta \sim 120°$ (Fig. 3c, d), which is consistent with previous findings in similar environments[11]. To reach a larger span of off-plane orientations, micrometric size spherical silica beads were then coated with DPPC/Chol lipid membranes labeled with NR, providing a well-defined wide range of 3D mean orientations similarly as in refs. 8,11,15 (Fig. 3e). In membrane-coated beads, both azimuthal and tilt orientations show consistent behavior with a spherical object as represented in Fig. 3 f: while azimuthal orientations $\xi$ follow a radial trend around the sphere's center for all image planes, i.e. the dipoles point towards the normal to the sphere surface, tilt angles $\eta$ increase from the bottom to the sphere's equator (Fig. 3e) which is expected from a progressive orientation change towards the plane of the sample. This trend is also visible in Fig. 3g, where averaged $\eta$ values over hundreds of single-molecule detections are represented as a function of their distance to the center of the bead image. The measured $\eta$ range does not span the whole (0–90°) magnitude, which can be explained by the tilt angle of NR with respect to the membrane normal direction, by the reduced $\eta$ range reachable in 4polar3D, and by the visible heterogeneity of the NR orientation behaviors on the sphere. Finally, the wobble value is similar to the SLB environment, albeit with a higher heterogeneity, which can be attributed to the less controlled quality of the lipid membrane on the sphere (Fig. 3c). Overall, these measurements confirm that 4polar3D SMOLM performs well in lipid membrane model systems.

## Imaging complex 3D actin filament meshworks in cells by 4polar3D SMOLM

Next, we applied the 4polar3D method to the imaging of the organization of dense actin filaments in 3D in cells. The organization of actin filaments is typically depicted as a 2D network in cells, although it is known to be fundamentally 3D in many regions of the cells, as shown by recent cryo-electron tomography works[36].

We first focused on the lamellipodium of migrating B16 melanoma cells (Fig. 4), a thin and dense actin filament network which expands a few micrometers from the cell leading edge[37]. In this cell area, the actin filaments' density is so large that deciphering their organization using SMLM imaging is a challenge[38]. Electron microscopy (EM) has provided evidence of 2D branched actin filament organizations at the lamellipodium edge, the branching arrangement governed by the Arp2/3 complex[37,39–41], with some degree of heterogeneity in the way filaments arrange[42]. This network has been observed in SMOLM using the 2D version of 4polar[25], yet direct quantification of 3D-tilted actin was not accessible. The 3D architecture of actin filaments in lamellipodia is much less studied. Recent cryo-electron tomography supported by modeling mentioned the

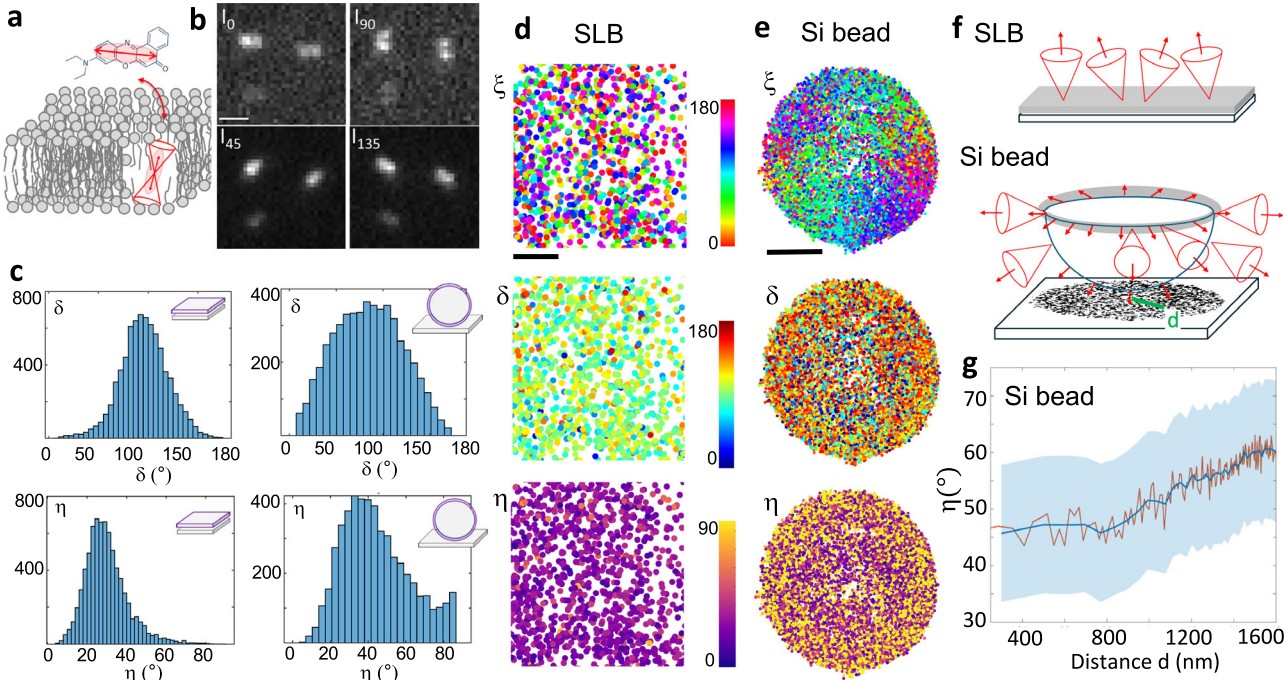

**Fig. 3 | 4polar3D in lipid samples. a** Schematic representation of a lipid bilayer labeled with transiently bound Nile Red (NR) molecules. **b** 4polar3D PSFs from single NR molecules, measured in a Single Lipid Bilayer (SLB) made of DPPC(60):Chol(40) mixtures, resulting from a sum over 20 images (integration time/image 100 ms). **c** Histograms of δ and η measured in SLB (left) as well as micrometric silica spheres (right) covered with lipid single bilayers of DPPC(60):Chol(40) mixtures, labeled with NR. The silica spheres data are reconstructed from the measurement of five different planes acquired by varying the tilt angles of the illumination, leading to an axial distance between the planes of about 600 nm from the sphere bottom to close to its equator. The data from three different spheres are plotted together. **d**, **e** 4polar3D maps of measured angular parameters (one marker is one molecule): ξ (top), δ (middle) and η (bottom) for **d** SLB and **e** three silica micro-spheres (Si beads, for which the five different measured planes are combined in one image). **f** Schematic representation of the SLB and Si beads samples, together with the direction of molecular dipoles (red arrows) within their wobbling cone. The resulting single molecule detections from the Si beads measurements are represented as black markers together with their distance d to the bead image center. **g** Averaged η values obtained from the three Si beads, as a function of the distance d (see **f**). Red line: average over 200 detections, blue line: average over 800 detections, for which the standard deviation is also represented as a shaded blue area. Scale bars: 1 µm (**b**), 100 nm (**d**), 1 µm (**e**).

existence of an actin network in 3D[43], however it is unclear if the Arp2/3 branching mechanism is limited to a 2D plane.

Using 4polar3D, we investigated both 3D orientation and wobble in areas where the actin filament density is high. We imaged fixed B16 cells labeled with the phalloidin conjugate AF568, which has been shown to be oriented along the actin filament direction, with a wobble of about 100°[25]. This wobble value is non negligible, but it is expected to allow the estimation of 3D mean orientations with high precision and a reduced bias on the off-plane angle η, as seen above. Figure 4a shows a typical 4polar3D cell map of η, ξ and δ values. This whole-cell actin 4polar3D image contains 2.4 million of molecular detections obtained from a stack of 60,000 images, for which the processing time, once the detection step is done, takes about 24 s on a standard workstation. This time is a thousand times faster than for PSF engineering methods, which require a minimization over multiple pixels and angular parameters[26].

Figure 4a shows that in the lamellipodium within 1-2 µm from the cell edge, ξ spans a large range of values, which confirms the complex meshwork formed in this region. Right behind the edge, a diversity of in-plane and off-plane populations can be observed, while ventral stress fibers, located at the cell center, are clearly in-plane and directional (Fig. 4a, b). Distributions of ξ in zoomed regions close to the cell edge reveal the existence of bimodal distributions of actin filament orientations as previously observed (Fig. 4c), which is consistent with a branched actin filament behavior promoted by Arp2/3[43]. These bimodal distributions vary along the cell contour, depicting trends with sometimes non-symmetric or broader, less well-defined distributions (Supplementary Fig. S10). Such variety of behavior has been pointed

out in EM studies[43] to be attributed to the local protruding or resting state of the cell edge[42,44]. A few micrometers away from the cell edge, in the transition zone between the lamellipodium and the lamella, ξ changes its behavior drastically, with actin filaments following the cell contour tangentially (Fig. 4a–c). This is in line with EM observations of long, linear bundles of actin filaments in this region[40,43,45]. In contrast, ventral stress fibers (SF) exhibit narrow and directional ξ distributions as expected (Fig. 4a–c). Remarkably, off-plane tilted 3D oriented actin filaments are predominantly found at the cell edge (Fig. 4d), with a progressive increase of the η values when moving from the cell edge to the transition zone, while stress fibers exhibit predominantly in-plane actin filaments (Fig. 4e). We note that η does not strictly reach off-plane (η ~ 0°) or in-plane (η ~ 90°) values. This is partly due to the known bias at these extreme regions. Moreover, the anchorage of AF568 dipole to the actin filaments takes place with an offset angle of about 20° filament[25], which means that strictly vertical filaments would lead to η ~ 20°. When investigating relations between in-plane and off-plane behaviors in different ROIs, we noticed that at the cell edge, the ξ bimodal distribution is more pronounced for in-plane molecules (η > 60°) than off-plane molecules (η < 40°), which depict less contrasted distributions (Fig. 4f, Supplementary Fig. S10). In contrast, distributions found in the transition zone or in SFs do not vary significantly when separating these populations. This behavior is also seen when averaging multiple ROIs (Supplementary Figs. S11, S12). Note that at extreme off-plane orientations, a higher error is expected for the estimation of ξ (Fig. 2d), which could contribute to the higher ξ disorder seen for off-plane filaments at the cell edge. This disorder is nevertheless still observed when discarding η values for which the

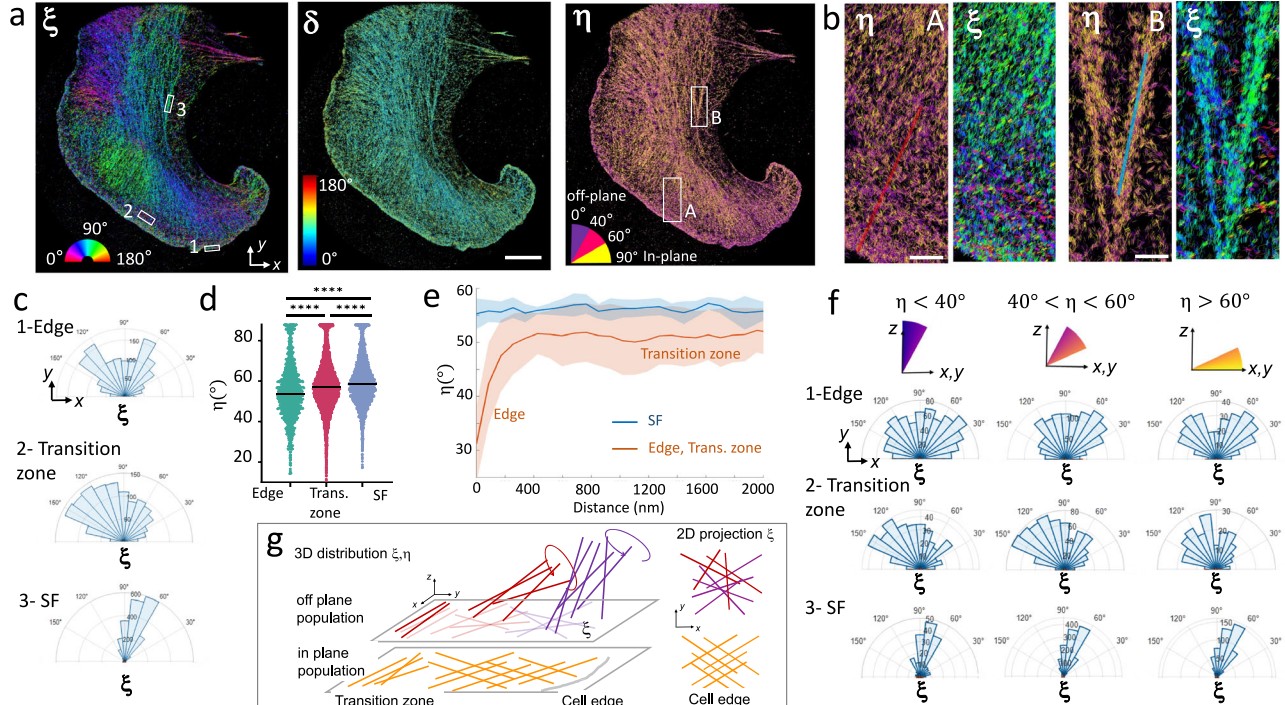

**Fig. 4 | 4polar3D imaging of 3D actin filaments organization in lamellipodia.**
**a** 4polar3D images on a fixed B16 cell labeled with phalloidin-AF568, displaying ξ (left), δ (middle) and η (right). η is shown using a binned colormap with indicated angular sectors. ξ, η, δ are encoded as colored lines whose orientation is along ξ for each localization. **b** zoomed η and ξ images of two different regions (shown as white squares A, B in (**a**)-right), illustrating the cell edge (left) and stress fibers (right). **c** Polar histograms of ξ for the three ROIs 1,2,3 shown as rectangles in (**a**)-left, respectively at the cell edge (1), in the transition zone between the lamellipodium and the cell inner region (2) and in a stress fiber (SF) region (3). **d** η values measured in the three regions (edge, transition zone, SF) of (**c**) (n = 1 516, 5 697, 3 792 molecules for ROIs 1,2,3 respectively). Statistical test: two-sided unpaired *t* test with *p* values < 0.0001 (****) for all compared populations. **e** In red: dependence of η

values with respect to the distance to the edge. The line and the shaded area are respectively the average and the standard deviation of η, measured over 30 profiles of 2 μm length, taken along the cell contour (see red profile example in (**b**)-left). In blue: similar result for 2 μm-length profiles taken along SFs (total of 15 lines, see blue profile example in (**b**)-right). **f** Polar histograms of ξ for given populations of η values: off-plane (η < 40°), intermediate (40° < η < 60°), in-plane (η > 60°), in the three ROIs of the cell displayed in (**c**). **g** Schematic representation of actin filaments at the cell edge and transition zone. At the cell edge the filaments form a network with a 70° branching angle with populations lying in-plane (yellow) and off-plane (red, purple). Both 3D and corresponding 2D projected distributions are shown. In the whole figure, (x, y) is the sample plane and z is the axial direction. Scale bars: 5 μm (**a**), 800 nm (**b**).

error on ξ is high (Supplementary Fig. S12). The branched actin network of the cell edge therefore seems to correspond to a predominantly in-plane population made of a ξ bimodal distribution (Fig. 4g, yellow), co-existing with an off-plane population of more isotropic, less organized filaments (Fig. 4g, red and purple). This off-plane network allows more diversity of ξ values, parly due to the larger rotational degrees of freedom of branched filaments (Fig. 4g). This behavior has not been extensively quantified in EM studies[41,46], even though models do not exclude 3D branched networks[47]. Specific mechanisms occuring at the very edge of the cell might be at the origin of this 3D-oriented population, where membrane proteins such as capping proteins concentrate to favor the recruitment of complexes for nucleation and elongation of new, short actin filaments[37], ensuring crucial and dynamic remodeling[41,45] for cell protrusion[43,44]. In contrast, the turnover rate of actin is critical at the back, a few micrometers away from the edge with long actin filaments parallel to the cell membrane[37,41,48]. The behavior of actin bundles at the transition zone and their connection to the branched population at the edge has been evidenced by EM[48], showing the branched actin is linked to these long actin filament subsets at large distances[37]. Overall, these distinct off-plane populations might correlate with the different actin layers reported in SMLM studies[38] or cryo-electron tomography[43]. The capacity of 4polar3D to detect and quantify the 3D organization of actin filaments in these regions emphasizes its significant added value for studies in whole cells and over large field of views. The wobble

angle δ follows a rather homogeneous behavior, which shows that the mobility of the phalloidin conjugate label is overall not affected by its environment (Fig. 4a). The capability to determine δ in a 3D sensitive manner is a strong asset compared to 4polar projected in 2D[25,26]. In 2D, the measurement of δ is strongly biased when fluorophores are tilted off-plane, as confirmed by Monte Carlo simulations (Supplementary Fig. S13). This is also visible at the edge of B16 cells, where the side-by-side comparison of 4polar imaging in its 3D and 2D versions shows a significant increase of measured δ for off-plane molecules in 4polar projected in 2D (Supplementary Fig. S13).

Finally, 4polar3D was applied to the imaging of 3D actin filament organization in podosomes, which form complex actin meshworks expanding in 3D, with a lateral size of about a micrometer. Podosomes are composed of an actin-rich core able to store elastic energy and generate protrusion forces[49,50], surrounded by an adhesion ring composed of integrins and adapter proteins coupled to the actin core with actin lateral filaments[51]. Imaging the organization of actin filaments in this complex network is challenging due to the diffraction-limited size of podosomes, whose core is about 250–300 nm wide. EM measurements have revealed that at the center of the core up to a thickness of about 200–400 nm[52], actin filaments are oriented off-plane. From the core center towards the border of the podosome, the actin network gets thinner and filaments gradualy become in-plane with a radial orientation around the center of the core. This spatial architecture was later corroborated by SMLM microscopy[50,53], although SMLM misses

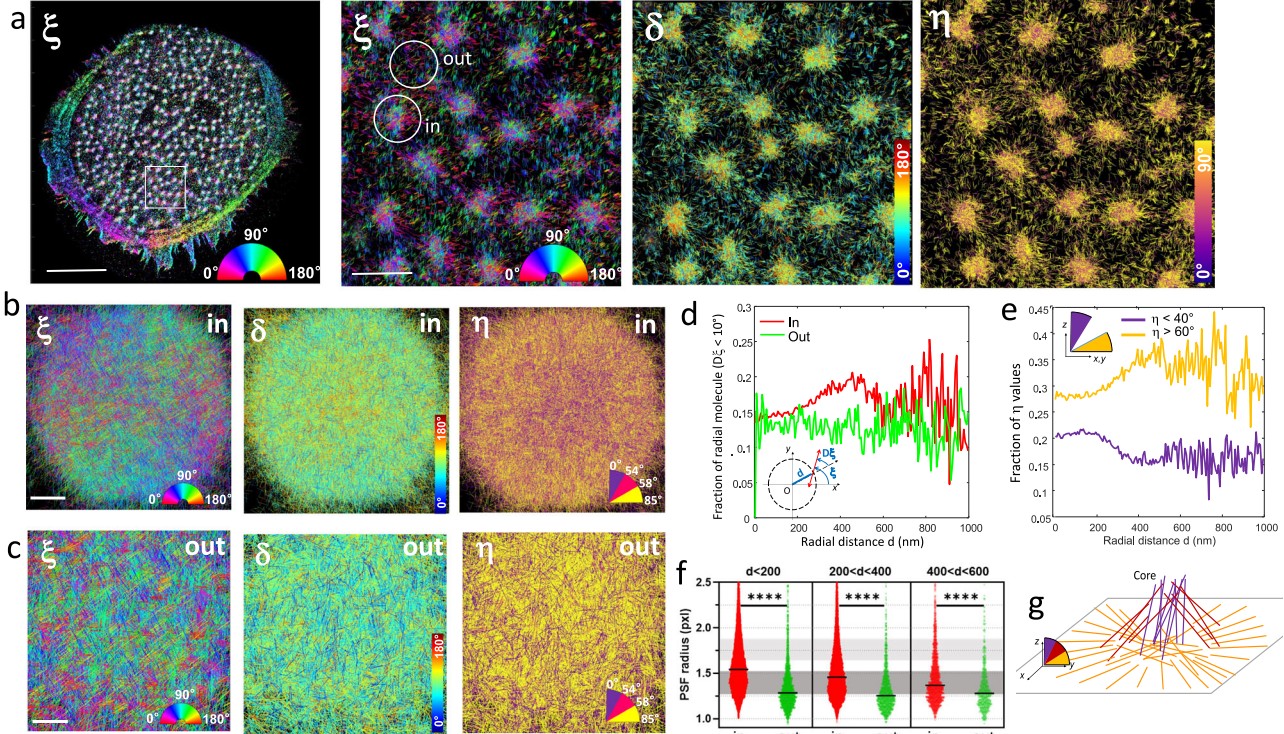

**Fig. 5 | 4polar3D imaging of 3D actin filaments organization in podosomes.**
Podosomes are imaged from Human primary monocyte-derived macrophages (hMDM) cells labeled with phalloidin-AF568. **a** In-plane orientation ξ diplayed for a whole cell (left), zoom on several podosomes (within the square shown on the left image) showing ξ, δ and η. The circles denote the chosen regions named 'in' (centered on a podosome) and 'out' (centered on the actin cloud surrounding podosomes). **b** 4Polar3D images produced from the superposition of 504 podosomes centered on their center of mass, analyzed from 6 cells (typically, 2000–7000 detections per podosome ROI). Each localization is a line oriented by its in-plane orientation angle (ξ), which color is coded by the corresponding parameter: ξ, δ, η. η is color coded by a binned value within the displayed angular ranges. **c** Similar 4Polar3D images produced from the superposition of 807 regions of similar sizes as in (**b**), outside of the podosomes, analyzed from 6 cells (typically, 100–500 detections per ROI). **d** Fraction of 'radial' molecules over the total number of localizations (defined as deviating from less than 10° from the radial direction to the ROI center), as a function of their distance to the ROI center (values are averaged over all localizations, within 10 nm distance steps), calculated over all localizations within 'in' (podosomes, red) and 'out' (actin cloud, green) ROIs displayed in (**b, c**). The fraction of radial molecules is defined by the proportion of the total localizations for which the radiality Dξ is between +/− 10°. The radiality Dξ is

defined as the angular difference between ξ and the angle carried by the vector from the ROI center to the localization. **e** Fraction of molecules (over the total number of localizations) within 'in' podosome regions, oriented off-plane (η < 40°) or in-plane (η > 60°), as a function of their distance to the ROI center. **f** Radius of the point spread function (PSF) measured on all localizations, for different ranges of distance d, for both 'in' and 'out' populations within ROIs displayed in (**b, c**). Populations 'in': $n = 15{,}350$ (d < 200 nm), 13,120 (200 nm < d < 400 nm), 1679 (400 nm < d < 600 nm); Populations 'out': $n$: 1839, 1090, 578. The gray ranges of values represent the expected PSF radius of theoretical PSFs for in-plane wobbling molecules (η = 90°, δ =100°, ξ random) (dark gray) and off-plane wobbling molecules (η = 0°, δ =100°, ξ random) (light gray). Statistical test: two-sided unpaired $t$ test with $p$ values < 0.0001 (****) for all compared populations. **g** Schematic representation of the organization of actin filaments in 3D in a podosome, depicting radial in-plane filaments (yellow) and tilted filaments of increasing off-plane angle η (red and purple). These off-plane filaments are located above the in-plane population at the core of the podosome. Note that for intermediate off-plane orientations, the two directions symmetric with respect to z are indistinguishable, therefore only one of them is represented, being the closest to published cryo-electron tomography data[36]. Scale bars = 5 μm (**a**, full cell), 1 μm (**a**, zooms), 200 nm (**b**), 100 nm (**c**).

the information on the 3D orientation of actin filaments in these structures, leaving open questions on the co-existence of ventral in-plane and off-plane actin networks[54] and on the organization of actin filaments between neighboring cores. Fig. 5a shows a typical 4polar3D image of a fixed Human primary monocyte-derived macrophages (hMDM) cell containing a large number of podosomes, labeled by phalloidin-AF568. While the actin fibers at the cell contour have well-defined orientations, podosomes contain a wide range of orientations, from which an in-plane radial behavior pointing out from the core center is perceptible, i.e. dipoles are oriented towards the normal to a circle centered at the podosome center position (zooms, Fig. 5a). The visible long range radial behavior is in line with previous EM imaging studies, which have shown that an actin network connects closely-associated podosome cores[54]. To visualize the actin architecture inside podosomes, averaged 4polar3D images over ~500 Podosomes (selected as 'in' regions in Fig. 5a and centered on their center of mass) are displayed in Fig. 5b. These images are made from typically 2 000–7000 single molecule detections/podosome. Figure 5b shows a

clear ξ radial behavior, while η decreases at the center of the podosomes, which is a signature of more off-plane filaments in the core region and gradually more in-plane filaments in the actin ring around the core. In contrast, regions between the podosomes (selected as 'out' in Fig. 5a and centered) display no specific ξ pattern, with dominating in-plane orientations (Fig. 5c). To quantify the radial actin behavior, we introduce the radiality Dξ, as the difference between the in-plane angle ξ and the radial angle of each single molecule, calculated from its position relative to the center coordinate of the region (Fig. 5d, inset). Figure 5d shows that in pododomes ('in' regions), the portion of radial molecules (quantified by the portion for which Dξ < 10°) is maximal at a distance $d$ ~ 400 nm from the podosome center, while regions taken between the podosomes ('out' regions) show no specific radial pattern. The distance $d$ ~ 400 nm, which defines a typical podosome size in EM[52], is also the one at which the in-plane actin population is the most predominant, as quantified by the η distributions in Fig. 5e. Figure 5e clearly shows that off-plane filaments are the most abundant at the center of the podosomes, in agreement with cryo-electron

tomography analyses[52]. Nevertheless, an important population of in-plane actin filaments co-exist with these off-plane filaments in the podosome core (Fig. 5b), which confirms the structural complexity of podosomes. A question that can be raised is how these different orientational populations are spatially distributed in height, over the podosome thickness. While 4polar3D does not permit to assess quantitative axial positions, the measured radius of the PSFs, when the image plane is set at the coverslip in total internal reflexion imaging conditions, permits to attribute the molecules to either 'in-focus' or 'out-of focus', keeping in mind that PSF radius also increases while η decreases (Supplementary Fig. S14). We observe that at the podosome center, in-plane/in-focus filaments (e.g. $\eta > 60°$ & PSF radius <1.4 pixels) co-exist with strongly off-plane/off-focus filaments (e.g. $\eta < 40°$ & PSF radius >2 pixels), while at the podosome border and outside podosomes, the PSFs appear to be mostly from in-focus, in-plane filaments (Fig. 5f). We note that correlations between height and off-plane orientations do not necessarily hold in the case of thick structures; indeed large ranges of PSF radii can be observed, for example, in thick stress fibers which nevertheless lie in plane (Supplementary Fig. S14). However, the maximal values of PSF radii observed in podosomes correspond to molecules lying typically off-plane and 400 nm above the coverslip, which is in line with the reported height of podosomes[50]. Our observations bring altogether new insights into the complex 3D architecture of actin networks in the podosome core, with off-plane filaments more predominantly positioned above a ventral structure, and in-plane filaments expanding radially beyond the podosome size as schematically represented in Fig. 5g.

## Discussion

In this work, we have demonstrated the use of an intensity-based method for single-molecule 3D orientation assessment, in addition to 2D localization. This approach, named 4polar3D, relies on a simple setup implementation, that is accessible in any image-splitting microscope and requires minimal calibration steps, without the need to estimate optical aberrations in the microscope detection path. The PSFs obtained in 4polar3D are less enlarged than in PSF engineering methods, allowing the exploration of high-density samples (i.e. with densely packed localizations) or dynamical situations that are challenging to PSF engineering such as single molecule tracking. The relatively simple PSF deformation also allows a straightforward estimation of their integrated intensities from SMLM-based estimation procedures, which makes signal processing both accessible and remarkably fast comparing to PSF engineering approaches. Notably, 4polar3D can be implemented in biological samples with high molecular density, which is typically the case in cells. The accuracy and precision of 4polar3D are within a few degrees for a wide range of molecular orientations in standard SNR scenarios, with limitations only at extreme orientational regimes, e.g. fixed/isotropic dipoles and totally in- or off-plane orientations. To mitigate the lower robustness of the method in these extreme regimes, we propose to implement angle binning, which is largely informative when one wants to differentiate orientational behaviors that are present in a given region of a complex sample.

We have showcased the capacity of 4polar3D to decipher the 3D organizations of actin filaments in complex actin meshworks in whole cells in lamellipodia and podosomes, over large fields of view. 4polar3D is particularly powerful for situations where statistics over a high number of regions are needed and without the need for physical sectioning or tomography, which makes this approach highly complementary to EM. Large fields of view are particularly useful for understanding transitions of orientational organizations across the cell surface at scales larger than a micrometer, and relate this organization to cell functions such as migration, protruding mechanisms and growth. Importantly, 4polar3D is fully compatible with co-localization studies of adapter proteins or actin-binding proteins, whose

nanometric scale localization is expected to strongly correlate with the structural 3D organization of actin filaments.

A significant added value of the way that 4polar3D is implemented is that it can potentially be extended to other modalities. Since 4polar3D is purely intensity-based, its extension to multicolor imaging is straightforward and requires only wavelength-associated calibration steps to accommodate for different chromatic polarization distortions occuring in microscope optics. 4polar3D is also extendable to the estimation of single molecules' axial position. It is for instance compatible with multiplane imaging methods, which are among the reliable approaches to achieve 3D localization over a large range of depths[55]. Adding an extra channel to capture the signal above the super critical angle could also give access to the distance of single molecules relative to the coverslip, using for instance an annular mirror[56].

Finally, the ratiometric nature of 4polar3D makes it applicable to ensemble imaging, which is a very strong asset compared to PSF engineering approaches. In this sense, it is fully compatible with the use of genetically encoded fluorescent-protein-based reporters for measurements of the 3D organization of actin filaments in living cells and tissues[57]. It provides a simple strategy compatible to wide field imaging, but also scanning modalities such as super resolution STED, MINFLUX, confocal and multiphoton scanning microscopy, thus potentially bringing an adequate solution to the current needs of polarized ensemble microscopy in 3D[58].

## Methods

### 4polar3D microscope

The 4polar3D microscope is a total internal reflection fluorescence (TIRF) imaging system based on a commercial inverted microscope (Eclipse Ti2-E, Nikon) with a custom-built TIRF excitation module and polarization sensitive imaging channels. A 561 nm continuous wave laser (Coherent) is used to excite the fluorophores, which is first expanded 10× (GBE10-A, Thorlabs) before being focused (AC254-400-A-ML, Thorlabs) at the back focal plane of the 100× NA 1.45 oil immersion objective (CFI SR HP Apo TIRF 100XC Oil, Nikon). The beam expander and focusing lens are mounted on translation stage, which enables switching between epi- and TIRF illumination. Before the beam is expanded, a pair of achromatic wave plates (AQWP05M-600 and AHWP05M-600, Thorlabs) are added to correct for polarization distortions that the laser can encounter from reflections, in particular on the dichroic mirror mirror (Di01-R405/488/561/635-25×36 Semrock). This minimizes the photoselection caused by the polarization distortions of the excitation beam. For measurements using AF568 and Nile-Red fluorophores, an emission filter (FF01-609/54-25 Semrock) was used.

In the 4polar3D imaging module at the right output port of the microscope, a non-polarizing beam splitter is used to split the fluorescence emission thereby creating two optical paths each having a 1× imaging relay using pairs of 150 mm achromatic lenses (AC254−150-A-ML and AC508-150-A-ML, Thorlabs). The resulting pixel size in the object plane is 130 nm. At each of these imaging paths, an iris diaphragm is placed at their respective conjugate back focal plane to control the effective numerical apertures. Along one of the imaging paths, a half-wave plate (AHWP05M-600, Thorlabs) is placed to rotate the polarization by 45° with respect to the other path. Wollaston prisms (Quartz Wollaston Polarizer 68−820, Edmund Optics) are then placed right after each of the conjugate back focal planes producing two sets of orthogonal linearly polarized imaging paths. The imaging paths are then directed using a mirror allowing all four imaging paths to be captured by the camera (ORCA-Fusion C14440-20UP, Hamamatsu, used in ultra-quiet mode with an imaging integration time of 100−185 ms.). Note that compared to PSF engineering methods, since the detected signal is reduced by the signal split and *NA* reductions, a compromise can be found to preserve a good signal to noise condition

by using integration times of typically 100 ms, or binning the PSF size down to only a few pixels. To compensate for the polarization distortions imposed by the dichroic mirror, a second identical dichroic mirror is mounted right after the exit port of the micrcoscope, in a vertical orientation that inverses the role of the p and s components of the fluorescence emission. This compensation dichroic is aligned to minimize the extinction for the 0/90 and 45/135 polarization channels, while observing unpolarized white light with a rotating polarizer in the emission path. Lastly, Bertrand lenses (AC254-100-A-ML and AC508-100-A-ML) are placed on flip mounts to image the back focal plane and adjust the numerical apertures of the system such that the high *NA* stays below the super critical angle regime and that the ratio between high and low *NAs* is roughly around 0.8. A more precise estimation of this ratio is contained into the full calibration of the setup (Supplementary Note 2). The microscope was also equipped with a three dimensional piezoelectric stage (U-780.DNS, Physik Instrumente) and the Nikon Perfect Focus System (PFS) to ensure minimal sample axial drift.

## Image processing

4Polar3D anaysis is run on custom MATLAB scripts (version R2024a, The MathWorks, Inc.) on a Dell Precision T5820 workstation (CPU, Intel Xeon W-2223 processor). The 4 channels images are first registered using as reference a fluorescent nanobeads samples. One of the polarized channels is used as a reference for the registration, for which affine transformations are defined for the three other images (Matlab Imaging Processing toolbox). Typical transformation parameters are given in Supplementary Note 2. A refined registration step is then applied directly on the single molecule data in the pairing process, similarly as in ref. 25, to refine the association of molecule's polarized projections. For nanobeads or single molecules, we apply a detection method inspired by Serge et al.[59], identical to the previously developed 4polar technique[25]. In this approach, detection parameters are obtained through a maximum-likelihood (ML) estimation and validated using a generalized likelihood ratio test (GLRT), which assesses whether the data inside a search window of 11 ×11 pixels is a true PSF peak or simply noise by comparing two Gaussian-noise hypotheses in which a probability of false alarm (for spurious peak detection) is fixed. The final PSF parameters (localization, intensity) are obtained in a second step, through a Gauss–Newton regression to minimize a least-squares fit based on a Gaussian PSF. During the detection stage, the PSF center is restricted to integer pixel coordinates and the PSF radius is kept fixed, whereas during the parameter estimation step, a sub-pixel precision Is used. Note that the detection step could be further improved and accelerated using a pure probability-based approach on a graphics processing unit (GPU), such as developed in ref. 60. In the present work, the main computational bottleneck remains, however, the fitting step for the localization and intensity parameters' estimation. Although additional acceleration strategies exist, using a 1D symmetric Gaussian PSF already provides a substantially faster solution than more complex PSF descriptions.

Even though the 1D symmetric Gaussian model used is known to be not adapted to polarized PSFs due to their distinct orientation-sensitive shapes, especially for fixed dipoles (see Supplementary Fig. S1), this is satisfactory for the detection step, as tested in Monte Carlo simulations (see Supplementary Note 4). Molecules are then coupled between two registered images (0–90) and (45–135) by selecting the nearest neighbor to the expected position at a vector distance from the reference image estimated from the registration step, within a distance tolerance corresponding to the localization precision. Then the four identified detections are associated. Computing the Fisher matrix and the associated Cramér Rao bound allows to estimate the theoretical lowest error expected for each PSF parameter as detailed in ref. 59, in particular the localization precision. This quantity is computed for all four polarization projections.

Once molecules are detected along the four channels, their parameters (2D localization, background level, PSF radius) are determined. Their intensities are estimated by PSF integration, using a PSF fit based on a Gaussian shape function (Gauss–Newton regression and minimization of a least squares analysis), a symmetric Gaussian function, or a box integration approach (see Supplementary Note 4). From the four intensities obtained along the four polarized channels, the parameters ($\eta,\xi,\delta$) are estimated for each molecule using the approach described in Supplementary Note 1. Before the final image representation, lateral drift correction is performed using the Thunder-STORM plugin on ImageJ (cross-correlation method with each bin containing 2000–5000 frames). A maximum averaged lateral drift of about 1 μm is observed over 1 h of measurement, while no axial drift is noticed. The final 4polar3D image depicts the parameters as sticks images, using one stick per molecule detected, whose orientation relative to the horizontal axis is $\xi$, and whose color is either encoding $\xi$, $\eta$ or $\delta$. For the display of results, $\eta$ and $\delta$ values that lead to physical indeterminations or high bias are discarded, such that detections whose parameters fall out of the confidence interval are excluded. This confidence interval is defined by $\eta$ within the range (5–88°), $\delta$ within the range (5–175°), and total detected intensity value above 1000 photons.

## Lamellipodia

B16-F1 cells (gift from Klemens Rottner, Technische Universität Braunschweig, Germany, originally from ATCC (CRL- 6323)), were cultured in DMEM (ThermoFisher Scientific, 41966-029) supplemented with 10% fetal bovine serum (PAA Laboratories, A15-102), 100 U/mL penicillin and 100 μg/mL streptomycin (Sigma, P4333) in a humidified incubator at 37 °C and 5% $CO_2$. The B16-F1 cell line was not authenticated and was not tested for mycoplasma contamination. H1.5 24 mm diameter (170 μm ± 5 μm) high precision coverslips (Marienfeld, 0117640) were sonicated in 70% Ethanol for 5 min and air-dried before being coated with mouse laminin (SIGMA L2020) for 1 h at room temperature (RT) and at a final laminin concentration of 25 μg/mL in coating buffer (50 mM Tris-HCl pH 7.5, 150 mM NaCl). B-16 cells were plated on the coated coverslips and allowed to spread overnight. To stimulate lamellipodia formation, cells were treated with aluminum fluoride for 15 min by adding $AlCl_3$ and NaF solutions to a final concentration of 50 μM AlCl3 and 30 mM NaF. Cells were fixed for 20 min with 0.25% glutaraldehyde and 4% formaldehyde in cytoskeleton buffer (10 mM MES pH 6.1, 150 mM NaCl, 5 mM EGTA, 5 mM $MgCl_2$, 5 mM glucose). The coverslips were washed twice with 1 mg/ml fresh $NaBH_4$ in PBS for 5 min each, followed by three washes with PBS for 5 min each. The cells were then incubated overnight with 0.5 μM AF568-Phalloidin in 0.1% saponin/10% BSA in PBS at 4 °C in a humidified chamber.

## Podosomes

Monocytes from healthy subjects (HS) were provided by Etablissement Français du Sang, Toulouse, France, under contract 21RB2025-028-R and 21RB2025-031-R. According to articles L12434 and R124361 of the French Public Health Code, the contract was approved by the French Ministry of Science and Technology (agreement number AC 2009921). Written informed consents were obtained from the donors before sample collection. Human peripheral blood mononuclear cells were isolated from the blood of healthy donors by centrifugation through Ficoll-Paque Plus (Cytiva), resuspended in cold phosphate-buffered saline (PBS) supplemented with 2 mM EDTA, 0.5% heat-inactivated Fetal Calf Serum (FCS) at pH 7.4 and monocytes were magnetically sorted with magnetic microbeads coupled with antibodies directed against CD14 (Miltenyi Biotec #130-050-201). Monocytes were then seeded on glass coverslips at $1.5 \times 10^6$ cells/well in six-well plates in RPMI 1640 (Gibco) without FCS. After 2 h at 37 °C in a humidified 5% $CO_2$ atmosphere, the medium was replaced by RPMI containing 10%

FCS and 20 ng/mL of Macrophage Colony-Stimulating Factor (M-CSF) (Miltenyi 130096489). Human primary monocyte-derived macrophages (hMDM) were harvested at day 7 using trypsin-EDTA (Fisher Scientific) and let adhere for 3 h in RPMI containing 10% FCS on 1.5 H precision glass coverslips (Marienfeld). Previously, the coverslips had been cleaned by a 15 min incubation at 80 °C in a RBS 35 solution (Carl Roth 9238, 1/500 diluted in milliQ water), were rinsed three times with milliQ water, let dry and sterilized at 175 °C for 120 min. For immunofluorescence, hMDM were fixed for 10 min at room temperature in a 3.7% (wt/vol) paraformaldehyde (Sigma Aldrich 158127) solution containing 0.25% glutaraldehyde (Electron Microscopy Sciences 16220) in Phosphate Buffer Saline (PBS) (Fisher Scientific). When indicated, before fixation, cells were mechanically unroofed at 37 °C using distilled water containing protease inhibitors (Thermo Scientific™ 87786) and 10 µg/mL phalloidin (Sigma-Aldrich P2141). After fixation, quenching of free aldehyde groups was performed by treatment with 50 mM ammonium chloride and 1 mg/mL NaBH4 in PBS, followed by three washes with PBS for 5 min each. The cells were then incubated overnight with 0.5 µM AF568-Phalloidin in 0.1% saponin/10% BSA in PBS at 4 °C in a humidified chamber.

### Nanobeads calibration sample

Fluorescent nanobeads (yellow-green Carboxylate-Modified FluoSpheres 0.1 µm, ThermoFisher Scientific F8803) are sonicated for 2 min, diluted in milliQ water by $10^5$ times and sonicated again for 2 min. Imaging spacers from Merck (GBL654004) are stuck on plasma-cleaned 1.5 H coverslips. 25 µl bead solution is placed in the well and allowed to dry completely. 20 µl milliQ water is added and a glass slide is stuck on the spacer. The sample is sealed with nail polish.

### STORM imaging buffer

The final composition of the buffer for 4polar-STORM measurements was 100 mM Tris-HCl pH 8, 10% w/v glucose, 5 U/mL pyranose oxidase (POD), 400 U/mL catalase, 50 mM β-mercaptoethylamine (β-MEA), 1 mM ascorbic acid, 1 mM methyl viologen, and 2 mM cyclooctatetraene (COT). D-(+)-glucose was from Fisher Chemical (G/0500/60). POD was from Sigma (P4234-250UN), bovine liver catalase from Calbiochem/Merck Millipore (219001-5MU), β-MEA from Sigma (30070), L-ascorbic acid from Sigma (A7506), methyl viologen from Sigma (856177), and COT from Sigma (138924). Glucose was stored as a 40% w/v solution at 4 °C. POD was dissolved in GOD buffer (24 mM PIPES pH 6.8, 4 mM MgCl2, 2 mM EGTA) to yield 400 U/mL, and an equal volume of glycerol was added to yield a final 200 U/mL in 1:1 glycerol:GOD buffer; aliquots were stored at −20 °C. Catalase was dissolved in GOD buffer to yield 10 mg/mL, and an equal volume of glycerol was added to yield a final 5 mg/mL (230 U/µL) of catalase in 1:1 glycerol:GOD buffer; aliquots were stored at −20 °C. β-MEA was stored as ~77 mg powder aliquots at −20 °C; right before use, an aliquot was dissolved with the appropriate amount of 360 mM HCl to yield a 1 M β-MEA solution. Ascorbic acid was always prepared right before use at 100 mM in water. Methyl viologen was stored as a 500 mM solution in water at 4 °C. COT was prepared at 200 mM in DMSO and aliquots stored at −20 °C. After mixing all components to yield the final buffer composition, the buffer was clarified by centrifugation for 2 min at $16,100 \times g$, and the supernatant kept on ice for 15 min before use. Freshly prepared STORM buffer was typically used within a day.

### Supported lipid bilayers

Supported lipid bilayers (SLB) were made following a solvent-assisted method[61]. Phospholipid stocks of 1,2-dipalmitoyl-sn-glycero-3-phosphocholine (DPPC) and cholesterol were purchased from Avanti Polar Lipids (Birmingham, AL, USA). In short, glass coverslips were prepared by treating them with piranha acid (3 parts of $H_2SO_4$ and 1 part of 30 wt % $H_2O_2$) for 10 min and then stored in pure water up to a month. This ensured that the glass surface is both hydrophilic and clean. Before

SLB production, coverslips were exposed to oxigen plasma for 5 min and then a microfluidic chamber (Sticky-SlideVI, Ibidi) was stick on it and the channels filled with water to avoid exposure with the air. Pharmed BPT and teflon tubes were used to input isopropanol in the selected channel at 100 µL/min for 10 min followed by the input of the selected lipid mixture concentrated at 0.2 g/L at 50 µL/min for 3 min. The flow was stopped for 6 min to allow for an incubation period prior to pump Tris-NaCl buffer (10 mM Tris and 150 mM NaCl, pH = 7.5) at 70 µL/min for 10 min.

### Silica beads coated with lipid bilayers

Silica beads of 3 µm diameter were purchased from Kisker Biotech GmbH & Co. KG.1,2-dipalmitoyl-sn-glycero-3-phosphocholine (DPPC) and cholesterol were purchased from Avanti Polar Lipids. Chloroform was purchased from VWR chemicals. All other reagents were purchased from Sigma-Aldrich. The method for the preparation of silica beads coated with lipid bilayers is based on reference[26]. Silica bead stock at 50 mg/ml was diluted to prepare a working concentration of 2 mg/ml. To prepare the 2 mg/mL bead solution, the bead stock was vortexed for 10 s and centrifuged at $16,450 \times g$ for 5 min, and the resulting pellet was resuspended in the TRIS buffer (20 mM TRIS and 50 mM KCl, pH 7.5). The vortexing and centrifugation processes were repeated 3 times. The resulting bead solution was kept in a water bath at 70 °C for 15 min, then supplemented with 20 mM of $CaCl_2$, vortexed for 30 s and kept in the water bath at 70 °C for further 15 min. 60 mole% DPPC and 40 mole% cholesterol were dissolved in chloroform in a glass vial. The chloroform was then evaporated under a gentle flow of nitrogen gas, resulting in a thin lipid film on the vial. The lipid film was kept overnight in a vacuum chamber to remove any excess residue of organic solvent. Next, the lipid film was rehydrated at a concentration of 1.4 mg/ml in the TRIS buffer, which was preheated to 70 °C in a water bath. The vial was sealed to prevent evaporation. The lipid suspension was incubated in a water bath at 70 °C for 30 min. During this incubation period, the lipid suspension was vortexed every 5 min. The hydrated lipid film was then sonicated in a bath sonicator at 70 °C for 30 min until the solution became completely transparent, indicating the formation of small vesicles. Immediately after the preparation of the small vesicles, the vesicle solution and the bead solution (at 2 mg/ mL) were mixed in a volume ratio of 1:1, and vortexed to mix them well. To allow the vesicles to fuse with each other and form lipid bilayers on the beads, the vesicle-bead mixture was incubated in a water bath at 70 °C for 1 h, while being vortexed every 10 min during this incubation time to facilitate complete coating of the beads with lipid bilayers. The resulting beads coated with lipid bilayers were allowed to cool slowly to room temperature for 1 h, while being vortexed every 15 min during the incubation period. To remove excess vesicles from the coated bead solution, the solution was centrifuged at $2800 \times g$ for 5 min at room temperature. The supernatant was discarded, and the pellet was replenished with the TRIS buffer and vortexed well. This procedure was repeated 5 times. The bilayer-coated beads were stored at 4 °C for a maximum of 2 weeks for experiments. For imaging, #1.5 coverslips were coated with 0.01% PLL for 20 min and washed thrice with Tris buffer. The SLB-coated bead solution was vortexed and mixed in Tris Buffer at 1:4 volume ratio. Nile Red is added to the solution at a final concentration of 0.75 nM. The mixture is vortexed again and added to the coverslip for observation.

### Reporting summary

Further information on research design is available in the Nature Portfolio Reporting Summary linked to this article.

## Data availability

The 4polar3D raw image stacks generated in this study are available upon request from the corresponding author (due to the large size of the files, 150-250 GB). A subset of data (2 GB), as well as the processed

orientation/detection parameters from single-molecule data generated in this study are available in the Source Data file (https://doi.org/10.6084/m9.figshare.28890470). Processed data for example ROIs generated in this study are available for download at https://github.com/CessVala/4polar3D_SMOLM (TestData folder), with explanations provided in the README.md file.

## Code availability

The MATLAB code used to analyze the data (The MathWorks, Inc. Recommended version: R2020a or newer) is available on GitHub at https://github.com/CessVala/4polar3D_SMOLM (https://doi.org/10.5281/zenodo.18663231), which includes a manual and installation instructions. The Python code used for the Monte Carlo simulation (Python Software Foundation. Recommended version, Python 3.8 or newer) is available on GitHub at https://github.com/CessVala/4polar3D_SMOLM_simulation (https://doi.org/10.5281/zenodo.18663511).

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

## Acknowledgements

The authors warmly thank Simli Dey and Feng-Ching Tsai (Institut Curie, Paris) for the preparation of the Silica beads coated with lipid bilayers. This research has received funding from the France 2030 investment plan managed by the PIA France 2030 program IDEC Equipex+ grant (ANR-21-ESRE-0002) and the Initiative d'Excellence d'Aix-Marseille Uni-versité (S.B.) - A*MIDEX Institutes Cancer et Immunologie (AMX-19-IET-001) and Marseille Imaging (C.S.S.-K.). This work is also funded by the ANR grants 3DPolariSR (ANR-20-CE42-0003) (S.B.) and SETIPSS (ANR-22-CE13-0039) (S.B., M.V.), the « Investissements d'Avenir » program managed by the ANR (ANR-16-CONV-0001) (S.B.), France BioImaging national infrastructure ANR-10-INBS-04-07 (S.B.), and from CNRS (S.B.). This work was also funded by the Chinese Academy of Sciences Pre-sident's International Fellowship Initiative grant 2024FSB0003 (C.A.V.-C.).

## Author contributions

C.S.S.-K. and M.S. co-developed the experimental setup, performed experiments and wrote data analysis codes. C.A.V.-C. developed the data processing and Monte Carlo simulation codes. L.A.A.-C. developed simulations on CRLB calculations. V.C. developed the initial version of the experimental setup. J.R.B and R.P prepared the cells dedicated to podosomes studies. M.M. supervised the cells preparations. M.A.A. contributed to data analysis decisions. S.B. conceived the project, developed simulations and wrote the manu-script. All authors contributed to feedbacks and edits on the manuscript.

## Competing interests

The 4polar3D microscope was invented by S.B. and V.C. and is covered by US patent 11927737 (2022), which was filed by and assigned to Uni-versité Aix Marseille, Centre national de la recherche scientifique and Centrale Marseille.
