## [Transparent Peer Review file · Nature Communications]

4polar3D single molecule imaging of 3D orientation in dense actin networks using ratiometric polarization splitting

Corresponding Author: Dr Sophie Brasselet

Version 0:

Reviewer comments:

Reviewer #1

(Remarks to the Author)

See attached.

Reviewer #2

(Remarks to the Author)

The manuscript by Senthil Kumar et al. presents a way to measure 3D orientation and 2D position of (fixed) fluorescent emitters with a relatively simple optical setup compared to earlier efforts.

Overall this is a very nice approach and useful for the field of single molecule orientation imaging. The approach can only do 2D imaging and is probably a bit worse in performance compared to more difficult optical setups and PSF fitting based approaches but has its right due to the simplicity.

Overall the manuscript could gain by 1) presenting a study on how aberrations influence the performance 2) a (short) comparison to earlier methods in terms of precision of position and orientation estimation to understand how much you loose by the simpler optical setup 3) rewording the abstract.

The title "Single molecule 3D orientation imaging.." Suggests 3D imaging, that is x,y,z position and orientation, while the paper only does x,y position (in plane) and orientation. Maybe adapt the title to "In plane single molecule 3D orientation ..." or similar

The abstract does not really go into the advantages of the method compared to other orientation estimations approaches, such as the easy setup and the robustness compared to aberrations. Maybe that could be added. In addition also only later the wobble estimation is mentioned, but this could also be added.

* L109. Make explicit that η is the polar angle and ξ is the in plane angle (now I need to look at the figure). Also indicate here already the angle range that is limited for the azimuthal angle l130.

* It might be good to state explicitly already in the abstract that the method in fact is not fully 3D as there remains an ambiguity in the ξ (from the in reality possible dipole orientations) that halves the Poincare sphere

L127 where in Sfig2 are other aberrations other than very small focal position variations treated? Here coma and astigmatism to lowest order at least must be checked.

L212 localization bias of 15 nm with an uncertainty of 20 nm quite a bit @5000 photons. The bias depends on the orientation and therefore will add a random offset similar to the localization uncertainty. This cannot be avoided as long as only the intensity is considered and no PSF fitting is done.

A study on how the bias would evolve for the number of photons to infinite would be good. Here 50,000 photons could be used in the simulation. This should be added to the discussion that the method might not be unbiased in the position - which is a downside to typical fitting based approaches.

L390 the discussion would benefit to compare the performance to earlier orientation and 2d/3d localization work. Now it is not clear what precision ones loses by intensity only measurement compared to PSF fitting while the optical simplicity of the setup is clear. It would be good to give some idea to the reader when it is needed to go into a more difficult setup and when a simpler is sufficient. Again the bias such be compared.

L414 It takes a while before it is clear that the polarization rotation introduced by the quad band dichroic is compensated by a second rotated mirror I430. Maybe add this earlier? Did you check that the dichroics are identical? same batch for the polarization distortion effects?

L440 on the registration side it would be nice to see some values of the rotation, shear and magnification changes between the channels from the affine matrix. Line 439 and later. The same for the drift.

*Sfig 3. A zoom in on the relevant part of eta and delta would be appreciated.

L170/171. It states that a GLRT is employed. The references are to older work of the authors, but in case the photon count is 1000+ with little background this very complicated and computational costly procedure is unnecessary [C.S. Smith, S. Stallinga, K.A. Lidke, B. Rieger, and D. Grünwald. Probability-based particle detection that enables threshold-free and robust in vivo single-molecule tracking. *Molecular Biology of the Cell*, 26(22):4057--4062, 2015.] How exactly was this implemented? If you need to compute the GRLT per pixel that would require a GPU implementation. A bit more information would be needed here.

Reviewer #3

(Remarks to the Author)

In their manuscript, Senthil Kumar and coauthors present a novel approach 4polar3D to localise single molecules at the same time as getting their orientation. Different than other recently developed methods, their approach does not depend on point-spread-function engineering and fitting to complex PSF shapes but on relatively straightforward image splitting. Orientation and location are determined from straightforward gaussian fits and intensity ratios. The key advantages of this approach are the (relative) simplicity of the setup, the simple and fast analysis pipeline and the ability to measure localisation and orientation at high densities (as demonstrated for f-actin in cells). In my view this is a highly innovative method, which in principle could be used by many others in the field. Orientation is unfortunately not used often enough, although it contains so much relevant biological insight! Overall the data and interpretation are convincing. And also generally well described, but it is rather complex matter, and I think here and there a bit more details could be provided and maybe the general reader could or should be taken a bit more by the hand. But overall, I think this is a very interesting manuscript that is a strong contender for Nature Comm!

Specific comments:

- fig 1 contains a lot of information, about bit more labels or text in the figure would be really helpful. I would add x,y,z to the cartoon insets in an and d. I would add "pupil plane" and "image plane" to middle and bottom figures of a.
- I was not completely convinced by what the authors want to show in 1a, in connection to the discussion of this figure 175-77. To me what was described is not so directly evident from the figure... Maybe some rescaling of the grey scales of only the regions within the red dashed rings could help???
- The authors uses irises to limit the effective NA of the objective. Fair enough. But how accurate is this? (How is this checked/determined?) And how dependent are the results on the actual NAs?
- 1123-125. It is not so clear where these ratios and the arctan and the norm come from. Is there physical logic why this works, or is this just a calculation that the authors propose and show that works?
- I was confused by Suppl. figure S4. What is sigma rho and rho (in C). In fact the deviation for the red curves between data and theory is quite large. This is not discussed. Should we be worried?
- I liked the use of the silica bead covered with a membrane to get a large distribution of angles. I think the authors should explain a bit better what is actually shown in the images in fig 3E. I understand that these are 5 image planes added. What z-positions? This could be made a bit more clear. Also the description: maybe help the reader to understand that what is seen is "consistent behavior with a spherical object" 1231.
- 1324-335: I found the description of the podosome data very hard to understand. E.g. " Figure 5b shows a clear ξ radial behavior" It would be helpful (at least to this reader) to help the reader a bit. What is the radial behaviour in the figure?
- for the biological data: would it be useful to provide a little cartoon summarising the results obtained for the actin filaments in podosomes and lamellopodia.

The text could be polished here and there:

- 126/27 : "measurements ... in a fast processing step". I guess that it is the 3d orientation determination or calculation from the measurement that is fast.
- 136 I guess mechanobiology is meant instead of mechabiology
- 138-41 I did not understand what the authors want to say...
- 167 distorTions. Maybe "optical aberrations" would be more clear.
- 178 achieve (instead of "achieve")
- 1317 "EM has evidenced a..." I would say something like "EM measurements have revealed that ..."
- 1341 abUndant

Version 1:

Reviewer comments:

Reviewer #1

(Remarks to the Author)

My previous comments have been satisfactorily addressed in the revised manuscript. I therefore recommend it for publication in Nature Communications. I list below only two very minor points to consider before final publication.

- Fig. 4a, panel 3 (eta image): I believe the labels "A" and "B" of the subregions are swapped relative to Fig. 4b.

- Fig. 5g, the schematic showing orientation trends relative to the center of the podosomes shows the intermediate- η molecules pointing tangent to the surface of the podosome I believe. This is at odds with the picture I had in my head in which the molecules tended to be oriented perpendicular to the podosome surface, which would otherwise coincide with the schematic for small η and large η . Since the method is invariant to 180 deg rotations about z, I suppose these two cases might be indistinguishable. Is there good reason to suspect one over the other?

Reviewer #2

(Remarks to the Author)

I am very happy with the extend and depth how the authors tackled the revision. Not only my, but also the others reviewers comments haven been taken up nicely. The paper improved quite a bit on the clarity and the few minor concerns and extensions have been addressed to my full satisfaction.

Reviewer #3

(Remarks to the Author)

In their revised version of the manuscript, the authors have carefully addressed my questions and (minor) concerns. In my view, this is a very interesting study that can be published in Nature Comm in its current form.

Response to reviewers – Manuscript NCOMMS-25-59202A

We would like to thank the reviewers for their thorough and constructive review of our work. We have addressed their comments below (blue), including the main changes here (blue-bold) and in the manuscript (red).

We have also added sub-figures in the main Figures as well as two Supplementary Figures, which we hope will improve the understanding of the points brought by the reviewers.

Best regards,
Sophie Brasselet, on behalf of the authors.

Reviewer #1

The authors present and demonstrate a new method for single-molecule orientation localization microscopy (SMOLM) based on polarization and aperture division called 4polar3D. The paper is in many ways a follow-up to the same group's development of 4polar2D as reported in Rimoli et al. Nat. Comm. 2022. The major improvement relative to the previous publication is the ability to estimate the out-of-plane component of the mean orientation as well as the full 3D wobble angle. Clearly access to the 3D orientation information is important, and thus the added novelty relative to the previous publication is evident. Experimental demonstrations of the technique applied to lamellopodia and podosomes are intriguing. Before acceptance for publication, however, a few criticisms and points-of-clarification should be addressed.

Criticisms

1. The method as designed wastes photons due to stopping down the NA (both to reject supercritical light and to achieve the NA difference between the channels). The cost of wasting photons may well be worth the gain due to simplicity of implementation and analysis, but a more forthcoming assessment should directly account for the implicit losses. What fraction of otherwise collected photons are lost?

We thank the reviewer for this important consideration. Indeed the reduction of the numerical aperture reduces the number of photons detected, by about a factor 2. We have estimated that 60% of the signal remains detected for the NA 1.3 channel (50% in the NA 1.1 channel), for an unpolarized emitter. Note that the NA 1.3 is imposed by the condition to stay below the SAF regime, which is necessary for samples in which the emitter's distance d to the coverslip is unknown. This is due to the mixture of polarizations occurring at the pupil plane above the critical angle, and to the sensitivity of intensity and polarization patterns to d . This is often discarded in SMOLM works, leading to biased information. The information on intensity losses has been added in the manuscript (Page 3) :

*« Typically, $NA_{high} \sim 1.3$ (corresponding to the critical angle aperture for a glass-water interface), **which corresponds to about 60% of the total signal remaining detected for an unpolarized emitter.** »*

How does this shift the CRB curves throughout the manuscript?

The CRB and Monte Carlo simulations in this work account for this loss of photons. Indeed the considered number of photons is given for the initial full NA of the objective, before any split/NA reduction, in order to be able to compare directly to other methods. This was not clear enough in the manuscript and it is now specified (Pages 5,6) : « total intensity 5 000 photons (**defined before any channel's split and NA reduction**) ». The CRB under Poisson noise conditions is known to follow a signal dependence of $1/\sqrt{N}$ with N the number of emitted photons. This is now mentioned in Page 5 of the manuscript :

“Note that under Poisson noise conditions, these standard deviations are expected to scale as $1/\sqrt{N}$, with N the number of detected photons.”

One price to pay for throwing away light is that one must integrate signal for longer to make up for it. To most, a significant increase in the measurement time would far outweigh an equivalent speedup in the analysis time.

Indeed, to reach sufficient precision the method requires a relatively high number of photons, therefore typical integration times used in the method are about 100-185ms for standard dyes. This is about a factor about 2 to 4 times longer than fast STORM/PALM methods, which is still way below the 1000's times faster analysis time compared to PSF engineering. Note that many SMOLM PSF engineering methods use similar integration times, due to the spread of the PSF over larger areas, leading to a lower signal per pixel. We have added a comment on this aspect in the Method section (Page 12) :

“Note that compared to PSF engineering methods, since the detected signal is reduced by the signal split and NA reductions, a compromise can be found to preserve a good signal to noise condition by using integration times of typically 100ms, or binning the PSF size down to only a few pixels.”

One could in principle avoid these losses by using annular mirrors to stop down the NA and then recycling the reflected light to another detector. It's certainly not worth building such an apparatus for the current publication, but a discussion along these lines may be worth including.

This is an interesting suggestion that would allow using lost photons in the localization estimation, for instance. However as pointed by the reviewer, the implementation of this method is not straightforward and would not bring significant added value in terms of orientation information. Nevertheless, such information could be used to estimate the distance of the emitter to the coverslip, providing the use of a full vectorial model taking its orientation into account. We have added this possibility in Page 11 (Discussion section) :

« Adding an extra channel to capture the signal above the super critical angle could also give access to the distance of single molecules relative to the coverslip, using for instance an annular mirror⁵⁶. »

2. One of the key biological conclusions drawn in this work is that (paraphrasing from lines 288-290) “the branched actin network of the cell edge seems to correspond to a predominantly in-plane population, co-existing with an off-plane population of more isotropic, less organized filaments”. The evidence cited for this conclusion is that bimodal ξ distributions appear in the in-plane populations but do not appear in the off-plane populations. However, couldn't this observation also be explained by the fact that your precision in estimate ξ is worse at small η (middle panel of Fig. 2d)? In other words, maybe you don't see bimodal distributions for small η because the measurement noise is blurring them. More work is needed to reject this alternative hypothesis.

This is a very good point, which we have addressed more carefully. In this work, we compare distributions belonging to $\eta < 40^\circ$ ('off-plane'), η between 40° and 60° (intermediate) and $\eta > 60^\circ$ ('in-plane'). The precision on ξ in the different categories was calculated from Monte Carlo simulations for roughly similar intensities: it is expected to be respectively $\sigma_\xi \sim 0^\circ-5^\circ$ ($\eta > 60^\circ$), $\sigma_\xi \sim 5-15^\circ$ (intermediate), and $\sigma_\xi \sim 15-55^\circ$ ($\eta < 40^\circ$). The most extreme error ($\sigma_\xi > 25^\circ$) occurs for $\eta < 30^\circ$. Therefore very off-plane molecules exhibits higher error on ξ . To see if the ξ distribution was affected by such possible loss of precision, we further filtered the η values so that very off-plane molecules were separated from less off-plane ones. This is shown in a new figure of the Supplementary Materials (Supplementary Fig. S12), where it is visible that there is no bimodal population occurring whatever the off plane range considered, even when the precision is high enough (i.e. below 10°). This is quite expected considering the behaviors for off plane filaments : indeed even under branching conditions, additional degrees of freedom in the rotation of branched filaments allow a wider variety of in-plane orientations to occur. This is shown in the schematic drawing added in Fig. 4 (Fig. 4g). A discussion on this effect is also added in Page 8 of the manuscript :

“Note that at extreme off-plane orientations, a higher error is expected for the estimation of ξ (Fig. 2d), which could contribute to the higher ξ disorder seen for off-plane filaments at the cell edge. This disorder is nevertheless still observed when discarding η values for which the error on ξ is high (Supplementary Figs. S11, S12). (...) This off-plane network allows more diversity of ξ values, partly due to the larger rotational degrees of freedom of branched filaments (Fig. 4g).”

Note that extreme values of η and δ are systematically removed from all analyzes on ξ , which thus discards very strong errors on the measurement of ξ . This information was initially unfortunately omitted, it is now added to the Method section (Page 13) :

« For the display of results, η and δ values that lead to physical indeterminations or high bias are discarded, such that detections whose parameters fall out of the confidence interval are excluded. This confidence interval is defined by η within the range ($5^\circ- 88^\circ$), δ within the range ($5^\circ- 175^\circ$), and total detected intensity value above 1 000 photons. »

Important technical points of clarification/improvements

1. Make sure that your definitions of which polarization channels correspond to NA_{high} vs. NA_{low} are consistent throughout the paper. Figure 1b-c indicates that the 0/90 polarization channels correspond to NA_{high}. If this is true, then shouldn't the ratio plotted in Fig. 1d be greater than 1?

Thanks for pointing this out, this is now corrected in the new version of Figure 1 which is consistent with the use of NA_{low} (0,90°) and NA_{high} (45,135°) through the whole work.

2. The color-coded images displayed throughout the figures are informative but sometimes hard to interpret. I found the projections in Figs. 5d and e very helpful for illustrating some of the important trends in the podosome images. Similar projections would be informative for the Si bead data as well. Something like a projection onto the contour of the cell edge might be fruitful for illustrating the important patterns extracted from the lamellopodia images.

We thank the reviewer for the pertinent suggestion : we have added such information for the results of the beads data (new Fig. 3g) and lamellipodia data (new Fig. 4e) as profiles, to evidence the spatial behavior of the h angle over relevant regions of the sample.

Additional minor points

1. Lines 85-86, the placement of the phrase "to within one sign" is confusing to me. Is this referring to the 180 degree ambiguity in determining the azimuth (not the polar angle)?

We apologize for the confusion, we modified this sentence to make the ambiguity clearer (Page 3) :

« ...determine the azimuth **angle ξ** (i.e., in-plane orientation **within the 0-180° range**), polar **angle η** (i.e. off-plane orientation), »

2. Line 103: "total energy radiated by a dipole is almost identical". I think you mean the total energy collected, not radiated. The total energy radiated seems like it would depend on orientation near the interface due to some anisotropic Purcell enhancement.

Indeed, sorry for the confusion, we modified 'radiated' into 'collected'.

3. From the inset in Fig. 1d it's difficult to tell whether Δ refers to the half- or full-angle of the cone.

Indeed, we added a double-headed arrow to indicate that d is the full angle.

4. Lines 131-132: "introducing an ambiguity between orientations symmetric with respect to the longitudinal axis z." Symmetric how? I think the answer is by 180 degree rotation.

Indeed, we have added the relevant information in this sentence (Page 4) : '**orientations symmetric by 180° rotation**'.

5. Lines 159-160 refers to "variances" of a few degrees, but you mean standard deviations. Variance would have units of degrees².

Indeed, we changed 'variance' into 'standard deviation' in two places in this paragraph.

6. Line 207 refers to Fig. 2h but I think 2f is meant.

This is modified, thanks for pointing this out.

7. Line 208: "the portion of molecules belonging to the wrong population is relatively low." A more quantitative statement is called for.

He have added the corresponding % in the corresponding sentence, Page 6 :

« by forming such binned populations of η , the portion of molecules belonging to the wrong population is ~~relatively low~~, is, for $\delta = 100^\circ$, 1 % (for the off-plane situation), 15% (intermediate) and 23% (in-plane). »

8. Line 305 refers to Fig. 4b but I believe they mean to refer to the middle panel of Fig. 4a.

This is now corrected.

9. Line 335 and Fig. 5 invoke the "radiality" denoted $\Delta \xi$. However, $\Delta \xi$ is already used to denote a different variable, the bias in estimating ξ . Use a different symbol.

Indeed, sorry about this : we modified the symbol denoting 'radiality', initially named ' $\Delta \xi$ ' into ' $D\xi$ ' through the whole manuscript and in Fig. 5.

10. Fig 1b: the aperture D1 appears to be off center, which is distracting/confusing given the points made in the text about the centrosymmetry of the apertures.

Thanks, this is now corrected.

11. Fig. 1c: the polarization directions indicated by the white double arrows are distractingly imprecise. The vertical arrow is clearly tilted. The two diagonal arrows are not mirror images of one another. See also my earlier point made about whether the high-NA/low-NA polarization channels might be mixed up—the contrast-adjusted images look like the 0/90 channels possess less light above background.

Thanks, this is now corrected.

12. Fig. 2e: in the eta histogram, an overlay of the distribution input into the simulation would help highlight the deviation.

The theoretical distribution of the input is now added in all histograms.

13. Fig. 4a-c: the insets overlaid in 4a and the panels in 4b and 4c all seem to have different sizes and aspect ratios from one another. Please ensure accuracy and consistency.

Sub-images initially in Figs. 4b,c are now replaced by Fig. 4b only, and their position and size is properly shown in Fig 4a (the corresponding regions are named A and B).

14. Fig 4b, left panel: there is a stray "0°" on the left part of the image.

Indeed, thanks for pointing this out, this is now removed.

15. Fig. 5a: add an inset to the far left panel to indicate the scene in the other three panels.

Thanks for pointing this out, this is now added (the corresponding regions are named 1,2,3).

Reviewer #2

The manuscript by Senthil Kumar et al. presents a way to measure 3D orientation and 2D position of (fixed) fluorescent emitters with a relatively simple optical setup compared to earlier efforts. Overall this is a very nice approach and useful for the field of single molecule orientation imaging. The approach can only do 2D imaging and is probably a bit worse in performance compared to more difficult optical setups and PSF fitting based approaches but has its right due to the simplicity.

Overall the manuscript could gain by 1) presenting a study on how aberrations influence the performance 2) a (short) comparison to earlier methods in terms of precision of position and orientation estimation to understand how much you lose by the simpler optical setup 3) rewording the abstract.

The title "Single molecule 3D orientation imaging.." Suggests 3D imaging, that is x,y,z position and orientation, while the paper only does x,y position (in plane) and orientation. Maybe adapt the title to "In plane single molecule 3D orientation ..." or similar.

We thank the reviewer for this point. The issue with incorporating 'in plane' in the title is that confusion might occur between 'in plane spatial' and 'in plane orientational'. To avoid adding extra words and making the title more complex, we changed the part 'Single molecule 3D orientation imaging' into 'Single molecule imaging of 3D orientation'. We suggest the new title:

« 4polar3D : Single molecule imaging of 3D orientation in dense actin networks using ratiometric polarization splitting »

The abstract does not really go into the advantages of the method compared to other orientation estimations approaches, such as the easy setup and the robustness compared to aberrations. Maybe that could be added. In addition also only later the wobble estimation is mentioned, but this could also be added.

We thank the reviewer for the suggestion, we have added more information to the abstract :

*« This strategy enables 3D orientation measurements of single molecules **within a 0-180° azimuthal range** in addition to **their angular range of fluctuations and their 2D localization, using a setup requiring minimal alignment complexity. It is moreover based on pure intensity-estimation, making data processing considerably faster than complex PSF shape analysis and relatively insensitive to geometrical aberrations.** »*

* L109. Make explicit that eta is the polar angle and xi is the in plane angle (now I need to look at the figure). Also indicate here already the angle range that is limited for the azimuthal angle l130.

This information is now added (Page 3) :

*« **splitting (e.g. NA filtering) to determine the azimuth angle ξ (i.e., in-plane orientation **within the 0-180° range**), polar angle η (i.e. off-plane orientation), and the **wobble angle δ of SMs (Fig. 1d, inset), from ratiometric intensity measurements.** »***

* It might be good to state explicitly already in the abstract that the method in fact is not fully 3D as there remains an ambiguity in the xi (from the in reality possible dipole orientations) that halves the Poincare sphere

This is now added in the abstract (see above), where the mention « **within a 0-180° azimuthal range** » has been added.

L127 where in Sfig2 are other aberrations other than very small focal position variations treated? Here coma and astigmatism to lowest order at least must be checked.

We thank the reviewer for this important point. In theory, the method is completely insensitive to geometrical aberrations since they bring a phase function to the pupil plane that does not affect its polarization content. However since aberrations affect the PSF shape and therefore potentially their intensity estimation, we have checked the effect of aberrations on the estimation of the 4polar3D parameters. We used Monte Carlo simulations for PSFs affected by coma and astigmatism, including defocus (spherical aberrations are expected to lead to similar results). The result is now added in the new Supplementary figures Figs. S8 and S9. If we exclude extreme situations (totally off-plane orientations), the graphs show that standard aberrations ($\lambda/4$ amplitude, i.e. $70m\lambda$ rms) introduce a degradation of the accuracy in δ , while the quality of the retrieval for the other parameters are not strongly affected. The result of this analysis is now added in Page 6 :

“Finally, Monte Carlo simulations were used to estimate how geometrical aberrations and defocus affect the performance of 4polar3D. While in theory, phase perturbations in the pupil plane do not have any incidence on the measured intensities, aberrations induce PSFs deformations that may lead to errors in the polarized channels’ intensity estimation under realistic noise conditions. Supplementary Figs. S8,S9 show that excluding extreme off-plane orientations, the precision and accuracy of orientation parameters is not strongly affected by the presence of standard aberrations, while the accuracy of δ is degraded by about 10 to 20°.”

L212 localization bias of 15 nm with an uncertainty of 20 nm quite a bit @5000 photons. The bias depends on the orientation and therefore will add a random offset similar to the localization uncertainty. This cannot be avoided as long as only the intensity is considered and no PSF fitting is done. A study on how the bias would evolve for the number of photons to infinite would be good. Here 50,000 photons could be used in the simulation. This should be added to the discussion that the method might not be unbiased in the position - which is a downside to typical fitting based approaches.

We thank the reviewer for this important point. It is indeed interesting to evaluate the intrinsic spatial localization error, which persists even in presence of high SNR conditions due to the nature of the intensity-based estimation. We have performed such simulations using 10 000 photons: the information is added to Fig. S7 and given in Page 6 of the manuscript :

“Note that the intrinsic localization bias due to the intensity-based nature of the estimation method is relatively low: at 10 000 photons (10 photons background/pixel), the expected (x, y) localization precision is around 6 nm for most of the orientations (10 nm for off-plane molecules) and the localization bias is about 3nm (8 nm for off-plane molecules) (Supplementary Figs. S7).”

L390 the discussion would benefit to compare the performance to earlier orientation and 2d/3d localization work. Now it is not clear what precision one loses by intensity only measurement compared to PSF fitting while the optical simplicity of the setup is clear. It would be good to give some idea to the reader when it is needed to go into a more difficult setup and when a simpler is sufficient. Again the bias should be compared.

Typically for PSF engineering approaches, precisions have been reached with values of a few degrees (2-10°) in the averaged 3D orientation (1000-2500 photons, 3-5 background photons/pixel), and about 10-20° for the wobble. We have added such information in Page 6 of the manuscript, referring to the most efficient published methods (references 7,8) :

“Overall, the estimation performance of 4Polar3D in orientation and spatial localization lies therefore in some aspects below the performance of the most efficient PSF engineering methods developed so far^{1,2}, for which precision and accuracy values have been found to reach a few degrees for all orientations and around 10° for wobble in similar signal to noise conditions^{7,8}. In contrast, the strong advantages of 4Polar3D lie in the significant simplification of its implementation, as well as its simple and fast data processing.”

L414 It takes a while before it is clear that the polarization rotation introduced by the quad band dichroic is compensated by a second rotated mirror L430. Maybe add this earlier? Did you check that the dichroics are identical? same batch for the polarization distortion effects?

The information is now added when the method is described in Page 3 of the manuscript :

« The polarization quality of both excitation and detection polarizations is ensured by compensation methods (see Methods). »

The identical nature of the dichroics, which were obtained together, was checked using a standard polarimeter. We have verified the considerable improvement of the polarization purity in the detection path, by inserting this second identical dichroic.

L440 on the registration side it would be nice to see some values of the rotation, shear and magnification changes between the channels from the affine matrix. Line 439 and later. The same for the drift.

The affine transformation parameters used to match different detection channels is now given in the Supplementary Materials (Supplementary Note 2). Regarding spatial drift, the highest level of lateral drift measured was typically 1mm over 1 hour of measurement, while no drift along the axial dimension was noticed thanks to the use of the autofocus system of the microscope. This information is now given in the Materials and Method section.

*SFig 3. A zoom in on the relevant part of eta and delta would be appreciated.

We have zoomed on the relevant η and δ data in Suppl. Fig. S3.

L170/171. It states that a GLRT is employed. The references are to older work of the authors, but in case the photon count is 1000+ with little background this very complicated and computational costly procedure is unnecessary [C.S. Smith, S. Stallinga, K.A. Lidke, B. Rieger, and D. Grünwald. Probability-based particle detection that enables threshold-free and robust in vivo single-molecule tracking. *Molecular Biology of the Cell*, 26(22):4057--4062, 2015.] How exactly was this implemented? If you need to compute the GRLT per pixel that would require a GPU implementation. A bit more information would be needed here.

The description of the algorithm was indeed short since it refers to existing works; we describe it now in more detail. The 4polar3D detection module employs indeed a GLRT approach as a validation step for PSF detections, as initially implemented in MTT (Sergé et al. *Nat. Methods*, 2008). As detailed in Smith et al. (*Mol. Biol. Cell*, 2015), more recent SM detection strategies can make a full use of GRLT and statistical tests to perform the SM detection step, which would necessitate a GPU-based implementation. Here, the GLRT calculation itself is not the main computational bottleneck and a standard workstation with CPU is used (as now specified in the Materials and Methods section). The most time-consuming component is the Gauss–Newton fitting step, which performs a least-squares optimization for every detected candidate. Thanks to the modular structure of 4polar3D, GPU-based implementations could be integrated in the future. We have completed the Materials and Methods section with more information in Page 12:

« In this approach, detection parameters are obtained through a maximum-likelihood (ML) estimation and validated using a generalized likelihood ratio test (GLRT), which assesses whether the data inside a search window of 11 x 11 pixels is a true PSF peak or simply noise by comparing two Gaussian-noise hypotheses in which a probability of false alarm (for

spurious peak detection) is fixed. The final PSF parameters (localization, intensity) are obtained in a second step, through a Gauss–Newton regression to minimize a least-squares fit based on a Gaussian PSF. During the detection stage, the PSF center is restricted to integer pixel coordinates and the PSF radius is kept fixed, whereas during the parameter estimation step, a sub-pixel precision is used. Note that the detection step could be further improved and accelerated using a pure probability-based approach on a graphics processing unit (GPU), such as developed in ⁶⁰. In the present work, the main computational bottleneck remains however the fitting step for the localization and intensity parameters' estimation. Although additional acceleration strategies exist, using a 1D symmetric Gaussian PSF already provides a substantially faster solution than more complex PSF descriptions."

Reviewer #3

In their manuscript, Senthil Kumar and coauthors present a novel approach 4polar3D to localise single molecules at the same time as getting their orientation. Different than other recently developed methods, their approach does not depend on point-spread-function engineering and fitting to complex PSF shapes but on relatively straightforward image splitting. Orientation and location are determined from straightforward gaussian fits and intensity ratios. The key advantages of this approach are the (relative) simplicity of the setup, the simple and fast analysis pipeline and the ability to measure localisation and orientation at high densities (as demonstrated for f-actin in cells). In my view this is a highly innovative method, which in principle could be used by many others in the field. Orientation is unfortunately not used often enough, although it contains so much relevant biological insight! Overall the data and interpretation are convincing. And also generally well described, but it is rather complex matter, and I think here and there a bit more details could be provided and maybe the general reader could or should be taken a bit more by the hand. But overall, I think this is a very interesting manuscript that is a strong contender for Nature Comm!

We thank the reviewer for these comments. We have addressed the points below, most of which involve providing additional explanations and clarifications. We hope that the revisions made to the manuscript and Supplementary Materials will help clarify these points.

Specific comments:

- fig 1 contains a lot of information, about bit more labels or text in the figure would be really helpful. I would add x,y,z to the cartoon insets in an and d. I would add "pupil plane" and "image plane" to middle and bottom figures of a.

We thank the reviewer for this comment, we have added helpful insets and information in Fig. 1 : 'pupil plane' and 'image plane' in a,c ; ' x,y ' in a ; name of the red ring ' SAF limit ' in a ; added pupil plane configurations in c.

- I was not completely convinced by what the authors want to show in 1a, in connection to the discussion of this figure l75-77. To me what was described is not so directly evident from the figure... Maybe some rescaling of the grey scales of only the regions within the red dashed rings could help???

We thank the reviewer for this comment, we have provided a more contrasted pupil plane image in Fig. 1c in order to make a clearer point on the relation between dipole orientation and energy balance and high and low NA in the pupil plane.

- The authors uses irises to limit the effective NA of the objective. Fair enough. But how accurate is this? (How is this checked/determined?) And how dependent are the results on the actual NAs?

The estimation of the effective NAs is first based on the direct imaging of the pupil plane thanks to Bertrand lenses (see Materials and Method section), where the SAF ring is used as a reference for the high NA channel. More importantly however, the factor between the high and low NAs is intrinsic to the calibration procedure (see Supplementary Note 2), which makes an estimate of the K matrix that accounts for both high and low NA channels. The obtained NAs are very close to those estimated by imaging the pupil plane of both paths of detection on the camera using Bertrand lenses (as mentioned in the Method section). We have added this information in the Supplementary Note 2 :

Note that this experimental determination of the matrix $\langle K \rangle$ embeds the estimation of the ratio between the high and low numerical aperture channels. It is therefore not necessary to estimate this factor separately.

As well as in the Materials and Method section :

Lastly, Bertrand lenses (AC254-100-A-ML and AC508-100-A-ML) are placed on flip mounts to image the back focal plane and adjust the numerical apertures of the system such that the high NA stays below the super critical angle regime and that the ratio between high and low NAs is roughly around 0.8. A more precise estimation of this ratio is contained into the full calibration of the setup (Supplementary Note 2).

- I123-125. It is not so clear where these ratios and the arctan and the norm come from. Is there physical logic why this works, or is this just a calculation that the authors propose and show that works?

The fact that the ratio between the two intensities $(I_0 + I_{90})$ and $(I_{45} + I_{135})$ depends on η is intrinsically due to the sensitivity of the pupil plane intensity pattern to the off-plane tilt angle of the dipole (Figs. 1a,c). The dependence of the other parameters $\text{atan}(P_{45}/P_0)$ and the norm $P_0^2 + P_{45}^2$ on ξ and δ come from a direct observation of Eq. S20 in Supplementary Note 1, where these parameters' sensitivities are visible from simple mathematical operations. These operations actually originate from a more fundamental effect: indeed in the paraxial regime (low NA) where orientations lie in 2D, $I_0 - I_{90} = m_{xx} - m_{yy}$, $I_{45} - I_{135} = 2m_{xy}$ and therefore the ratios of intensities $P_0 = (I_0 - I_{90})/(I_0 + I_{90})$ and $P_{45} = (I_{45} - I_{135})/(I_{45} + I_{135})$ are directly related to the normalized 2D Stokes parameters S_1/S_0 and S_2/S_0 , which are themselves the signature of the orientation of linearly polarized states in 2D (corresponding here to the orientation of a 2D dipole). In Stokes polarimetry, $\text{atan}(P_{45}/P_0)$ and $P_0^2 + P_{45}^2$ would thus correspond to $\text{atan}(S_2/S_1)$ and $(S_1^2 + S_2^2)/S_0^2$, related to polarization/dipole orientation and degree of polarization/dipole wobble. To explain this better, we have added a sentence in Page 4 of the manuscript :

« Second, we notice that in the paraxial regime, the ratios $P_0 = (I_0 - I_{90})/(I_0 + I_{90})$ and $P_{45} = (I_{45} - I_{135})/(I_{45} + I_{135})$ can be directly related to the normalized Stokes parameters, from which 2D orientation ξ and wobble δ can be determined through simple mathematical operations (Supplementary Note 1).»

and the reason of choice of these ratios is now more explicated in in Supplementary Note 1 :

“Note that in the paraxial regime in a perfect optical system with identical NAs for both channels, the $\langle K \rangle_{2D}$ matrix is considerably simplified such that $I_0 = m_{xx}$, $I_{90} = m_{yy}$, $I_{45} = I_{135} = 2m_{xy}$, therefore the moments are related to intensities with simple expressions: $P_0 = (I_0 - I_{90})/(I_0 + I_{90}) = (m_{xx} - m_{yy})/(m_{xx} + m_{yy})$, $P_{45} = (I_{45} - I_{135})/(I_{45} + I_{135}) = 2m_{xy}/(m_{xx} + m_{yy})$. The factors P_0 and P_{45} also correspond to the normalized 2D Stokes parameters S_1/S_0 and S_2/S_0 , which are themselves the signature of the orientation of linearly polarized states in 2D (corresponding here to the orientation of a 2D dipole). In Stokes polarimetry, the quantities used to retrieve the polarization direction (i.e. dipole orientation) and degree of polarization (i.e. dipole wobble) are respectively $\text{atan}(S_2/S_1) = \text{atan}(P_{45}/P_0)$ and $(S_1^2 + S_2^2)/S_0^2 = P_0^2 + P_{45}^2$.”

- I was confused by Suppl. figure S4. What is sigma rho and rho (in C). In fact the deviation for the red curves between data and theory is quite large. This is not discussed. Should we be worried?

The ρ variable is an error, it should be ξ , thanks for pointing this out, this graph is now corrected. Regarding the difference between experimental and CRLB curves, it is very common to obtain experimental data that are a few times the CRLB values. Indeed the CRLB is the smallest error to be expected from the measurement, not specifying which estimator is used. This is now included in the legend of Fig. S4 :

“Note that the difference in standard deviations between experimental and CRLB values can reach about 5° for some parameters: this is not surprising considering the fact that CRLB correspond to an optimal error value, while experimental values contain errors coming from the detection and the estimation process.”

- I liked the use of the silica bead covered with a membrane to get a large distribution of angles. I think the authors should explain a bit better what is actually shown in the images in fig 3E. I understand that these are 5 image planes added. What z-positions? This could be made a bit more clear. Also the description: maybe help the reader to understand that what is seen is "consistent behavior with a spherical object" l231.

Five different planes were measured from the equatorial plane (considered to be the plane of largest diameter) down to the plane close to the coverslip. Considering that the beads are around 3 μm diameter, this means about one plane every 600 nm. The depicted data are obtained by plotting the results from these five planes together. This information is now added in the legend of the figure. The description of the behavior of the dipoles around the sphere is now better explained in Page 7 of the manuscript, together with a schematic drawing added in Fig. 3:

« ...while azimuthal orientations ξ follow a radial trend around the sphere's center for all image planes (i.e. the dipoles point towards the normal to the sphere surface) »

- l324-335: I found the description of the podosome data very hard to understand. E.g. " Figure 5b shows a clear ξ radial behavior" It would be helpful (at least to this reader) to help the reader a bit. What is the radial behaviour in the figure?

Thanks for this remark, we have now better specified what a radial behavior means in Page 6 of the manuscript :

« from which an in-plane radial behavior pointing out from the core center is perceptible (zooms, Fig. 5b), i.e. dipoles are oriented towards the normal to a circle centered at the podosome center position. »

We have also added a cartoon displaying more clearly the actin organizations found in these data, which is now added in Fig. 5.

- for the biological data: would it be useful to provide a little cartoon summarising the results obtained for the actin filaments in podosomes and lamellopodia.

This is a very good point, we have added in Figs 4 and 5 such cartoons, which should hopefully help the reader understanding the outcome of our observations.

The text could be polished here and there:

- l26/27 : "measurements ... in a fast processing step". I guess that it is the 3d orientation determination or calculation from the measurement that is fast.

Indeed, thanks for pointing this out : this is now rephrased in the new version of the abstract :

« It is moreover based on pure intensity-estimation, making data processing considerably faster than complex PSF shape analysis... »

-l36 I guess mechanobiology is meant instead of mechabiology

Indeed thanks, it is now corrected.

-l38-41 I did not understand what the authors want to say...

« For example the orientation of actin filaments in the cell cytoskeleton, whose structure is typically inaccessible to standard single molecule localization microscopy (SMLM) due to their high molecular density, can be better probed using orientation-resolved techniques. »

The added value of orientation in addition of position is clearly found when the image cannot help guessing which orientations the actin network is made of, because of the high density of the

localizations in particular. We have modified the sentence in the introduction Page 2, to make this clearer :

« To probe the orientational organization of proteins in a cell, standard single molecule localization microscopy (SMLM) is often not enough. For instance actin filaments in the cell cytoskeleton are so dense that such organization is not visible within the packed collection of localizations. Therefore specific orientation-sensitive methods need to be developed. »

-l67 distortions. Maybe "optical aberrations" would be more clear.

Thanks, we have indeed replaced the term by *optical aberrations*.

-l78 achieve (instead of "achieve")

Thanks, this is now corrected.

-l317 "EM has evidenced a..." I would say something like "EM measurements have revealed that ..."

Thanks, this is now corrected.

-l341 abundant

Thanks, this is now corrected.

February, 15th 2026

Dear Editor,

We would like to thank the reviewers for their last comments on our revised manuscript. Please find below a point-by-point response (in blue text) to these comments. The modifications made to the manuscript are visible on the .doc corrected-manuscript version, in red.

Best regards

Sophie Brasselet, on behalf of the authors.

Point-by-point responses :

Reviewer #1 (Remarks to the Author):

My previous comments have been satisfactorily addressed in the revised manuscript. I therefore recommend it for publication in Nature Communications. I list below only two very minor points to consider before final publication.

We thank the reviewer for their positive comment.

- Fig. 4a, panel 3 (image): I believe the labels "A" and "B" of the subregions are swapped relative to Fig. 4b.

We thank the reviewer for noticing this, the labels have been corrected in Fig. 4.

- Fig. 5g, the schematic showing orientation trends relative to the center of the podosomes shows the intermediate- η molecules pointing tangent to the surface of the podosome I believe. This is at odds with the picture I had in my head in which the molecules tended to be oriented perpendicular to the podosome surface, which would otherwise coincide with the schematic for small η and large η . Since the method is invariant to 180 deg rotations about z, I suppose these two cases might be indistinguishable. Is there good reason to suspect one over the other?

The reviewer is perfectly right, there is no distinction between the intermediate filaments that would be tangent to or tilted away from the podosome surface. Nevertheless we chose to represent these filaments in a form that is close to what is known from cryo-electron tomography. This is now included in the new version of the manuscript, legend of figure 5 : « Note that for intermediate off-plane orientations, the two directions symmetric with respect to z are indistinguishable, therefore only one of them is represented, being the closest to published cryo-electron tomography data³⁶».

Reviewer #2 (Remarks to the Author):

I am very happy with the extend and depth how the authors tackled the revision. Not only my, but also the others reviewers comments haven been taken up nicely. The paper improved quite a bit on the clarity and the few minor concerns and extensions have been addressed to my full satisfaction.

We thank the reviewer for their positive comment.

Reviewer #3 (Remarks to the Author):

In their revised version of the manuscript, the authors have carefully addressed my questions and (minor) concerns. In my view, this is a very interesting study that can be published in Nature Comm in its current form.

We thank the reviewer for their positive comment.

The authors present and demonstrate a new method for single-molecule orientation localization microscopy (SMOLM) based on polarization and aperture division called 4polar3D. The paper is in many ways a follow-up to the same group's development of 4polar2D as reported in Rimoli et al. Nat. Comm. 2022. The major improvement relative to the previous publication is the ability to estimate the out-of-plane component of the mean orientation as well as the full 3D wobble angle. Clearly access to the 3D orientation information is important, and thus the added novelty relative to the previous publication is evident. Experimental demonstrations of the technique applied to lamellopodia and podosomes are intriguing. Before acceptance for publication, however, a few criticisms and points-of-clarification should be addressed.

Criticisms

1. The method as designed wastes photons due to stopping down the NA (both to reject supercritical light and to achieve the NA difference between the channels). The cost of wasting photons may well be worth the gain due to simplicity of implementation and analysis, but a more forthcoming assessment should directly account for the implicit losses. What fraction of otherwise collected photons are lost? How does this shift the CRB curves throughout the manuscript?

One price to pay for throwing away light is that one must integrate signal for longer to make up for it. To most, a significant increase in the measurement time would far outweigh an equivalent speedup in the analysis time.

One could in principle avoid these losses by using annular mirrors to stop down the NA and then recycling the reflected light to another detector. It's certainly not worth building such an apparatus for the current publication, but a discussion along these lines may be worth including.

2. One of the key biological conclusions drawn in this work is that (paraphrasing from lines 288-290) "the branched actin network of the cell edge seems to correspond to a predominantly in-plane population, co-existing with an off-plane population of more isotropic, less organized filaments". The evidence cited for this conclusion is that bimodal ξ distributions appear in the in-plane populations but do not appear in the off-plane populations. However, couldn't this observation also be explained by the fact that your precision in estimate ξ is worse at small η (middle panel of Fig. 2d)? In other words, maybe you don't see bimodal distributions for small η because the measurement noise is blurring them. More work is needed to reject this alternative hypothesis.

Important technical points of clarification/improvements

1. Make sure that your definitions of which polarization channels correspond to NA_{high} vs. NA_{low} are consistent throughout the paper. Figure 1b-c indicates that the 0/90 polarization channels correspond to NA_{high} . If this is true, then shouldn't the ratio plotted in Fig. 1d be greater than 1?

2. The color-coded images displayed throughout the figures are informative but sometimes hard to interpret. I found the projections in Figs. 5d and e very helpful for illustrating some of the important trends in the podosome images. Similar projections would be informative for the Si bead data as well. Something like a projection onto the contour of the cell edge might be fruitful for illustrating the important patterns extracted from the lamellopodia images.

Additional minor points

1. Lines 85-86, the placement of the phrase “to within one sign” is confusing to me. Is this referring to the 180 degree ambiguity in determining the azimuth (not the polar angle)?
2. Line 103: “total energy radiated by a dipole is almost identical”. I think you mean the total energy collected, not radiated. The total energy radiated seems like it would depend on orientation near the interface due to some anisotropic Purcell enhancement.
3. From the inset in Fig. 1d it’s difficult to tell whether Δ refers to the half- or full-angle of the cone.
4. Lines 131-132: “introducing an ambiguity between orientations symmetric with respect to the longitudinal axis z.” Symmetric how? I think the answer is by 180 degree rotation.
5. Lines 159-160 refers to “variances” of a few degrees, but you mean standard deviations. Variance would have units of degrees².
6. Line 207 refers to Fig. 2h but I think 2f is meant.
7. Line 208: “the portion of molecules belonging to the wrong population is relatively low.” A more quantitative statement is called for.
8. Line 305 refers to Fig. 4b but I believe they mean to refer to the middle panel of Fig. 4a.
9. Line 335 and Fig. 5 invoke the “radiality” denoted $\Delta \xi$. However, $\Delta \xi$ is already used to denote a different variable, the bias in estimating ξ . Use a different symbol.
10. Fig 1b: the aperture D1 appears to be off center, which is distracting/confusing given the points made in the text about the centrosymmetry of the apertures.
11. Fig. 1c: the polarization directions indicated by the white double arrows are distractingly imprecise. The vertical arrow is clearly tilted. The two diagonal arrows are not mirror images of one another. See also my earlier point made about whether the high-NA/low-NA polarization channels might be mixed up—the contrast-adjusted images look like the 0/90 channels possess less light above background.
12. Fig. 2e: in the eta histogram, an overlay of the distribution input into the simulation would help highlight the deviation.
13. Fig. 4a-c: the insets overlaid in 4a and the panels in 4b and 4c all seem to have different sizes and aspect ratios from one another. Please ensure accuracy and consistency.
14. Fig 4b, left panel: there is a stray “0°” on the left part of the image.
15. Fig. 5a: add an inset to the far left panel to indicate the scene in the other three panels.